# Quantifying cell transitions in *C. elegans* with data-fitted landscape models

**Elena Camacho-Aguilar** [1,2] *, **Aryeh Warmflash** [2,3], **David A. Rand** [1,4] *

**1** Mathematics Institute, University of Warwick, Coventry, United Kingdom, **2** Department of Biosciences, Rice University, Houston, Texas, United States of America, **3** Department of Bioengineering, Rice University, Houston, Texas, United States of America, **4** Zeeman Institute for Systems Biology & Infectious Disease Epidemiology Research, University of Warwick, Coventry, United Kingdom

* ec47@rice.edu (EC-A); d.a.rand@warwick.ac.uk (DAR)

## Abstract

Increasing interest has emerged in new mathematical approaches that simplify the study of complex differentiation processes by formalizing Waddington's landscape metaphor. However, a rational method to build these landscape models remains an open problem. Here we study vulval development in *C. elegans* by developing a framework based on Catastrophe Theory (CT) and approximate Bayesian computation (ABC) to build data-fitted landscape models. We first identify the candidate qualitative landscapes, and then use CT to build the simplest model consistent with the data, which we quantitatively fit using ABC. The resulting model suggests that the underlying mechanism is a quantifiable two-step decision controlled by EGF and Notch-Delta signals, where a non-vulval/vulval decision is followed by a bistable transition to the two vulval states. This new model fits a broad set of data and makes several novel predictions.

**Data Availability Statement:** All data and code are publicly available from the GitHub repository at: https://github.com/ecamacho90/VulvalDevelopment.

## Author summary

Standard models of cell differentiation focus on creating and analyzing gene regulatory networks (GRNs), which can be used to predict the evolution of a gene expression profile and determine stable states that correspond to cell fates. These models require a large number of parameters and variables, and the difficulty in constraining the parameters can reduce their predictive value. Further, model complexity often limits the ability to offer mechanistic insight. Recently there has been an increased interest in simplified models that mathematically formalize Waddington's landscape metaphor, focusing on cell fates and the transitions between them at the phenotypic level without reference to the underlying GRN. However, to date there is no general, systematic method to develop and fit landscape models to new biological problems. Here we present a methodology to formulate landscape models and fit them to quantitative data, and apply it to model the well-studied process of vulval development in the *C. elegans* worm. This model represents a qualitatively novel way of thinking about this process, reproduces a large quantity of existing data, and makes novel predictions. Moreover, we provide all necessary mathematical

**Funding:** This work was funded by the University of Warwick and Rice University, scholarship to ECA from EPSRC (1499350 - A New Mathematical Approach to Cell Differentiation Using Singularity Theory), grants to DAR from EPSRC (EP/P019811/1 - Mathematical Foundations of Information and Decisions in Dynamic Cell Signalling), and to AW from NSF (MCB-1553228), NIH (R01GM126122), Simons Foundation (511079), and Welch Foundation (C-2021). Some of this work was also performed at the KITP Santa Barbara and therefore this research was supported in part by the National Science Foundation under Grant No. NSF PHY-1748958, NIH Grant No. R25GM067110, and the Gordon and Betty Moore Foundation Grant No. 2919.01. The funders had no role in study design, data collection and analysis, decision to publish, or preparation of the manuscript.

**Competing interests:** The authors have declared that no competing interests exist.

background and implementation details as well as software that can serve as a resource for the broader community to apply in other contexts.

## Introduction

A key stage in the development of an organism is cell differentiation, in which unspecialized cells, called stem cells, become specialized ones depending on the signals that they receive. This is controlled by a very large network of interacting genes [1], the state of which defines the characteristics of the cell. However, this process is still not completely understood.

With the recent development of experimental techniques that allow us to obtain detailed quantitative information about the state of cells over time, new data analysis methods and mathematical models are required for the understanding of cell differentiation. A common approach to the modeling of stem cell differentiation is by means of gene regulatory network (GRN) models, which aim to define the possible differentiation states of a cell by its genetic expression profile. However, these models require a large number of parameters and variables which rapidly increases with the size of the network, complicating its analysis. Moreover, the complexity of such models often means they offer little mechanistic insight.

With this in mind, there has recently been an increasing interest in new kinds of mathematical models that formalize Waddington's epigenetic landscape metaphor [2], without reference to the underlying molecular network [3–6]. In this picture, a differentiating cell is represented as a marble rolling down a landscape of hills and valleys, encountering decision points between different lineages, and eventually settling in a valley that defines its cell fate. In particular, the models developed in [3, 5] reason directly on the phenotype, expressed in geometrical terms, focusing on the dynamics of the general process rather than on the deep molecular scale. These models contain the essence of the process that is necessary for its understanding, which is the structure of cell fates in the Waddington landscape and the effect that inducing signals have upon this landscape. In their formulation, the relevant dynamical systems are gradient-like, and the system's trajectories, which represent the developmental path of a cell, move downhill in this landscape until they reach a minimum representing the cell's state. Variations in the signals received by the cell change the underlying control parameters of the landscape and cause bifurcations to occur, allowing the cell's state to change by moving to another minimum.

In these previous models a function representing the landscape was formulated in an ad-hoc manner, however, and developing a general method to mathematically construct these landscape models remains an open problem. The key features of the models are the ways in which the attractors appear and disappear, i.e. the allowed bifurcations, and how the biological signals are mapped into the parameters representing the landscape. Catastrophe Theory (CT) and Dynamical Systems Theory provide very powerful tools for classifying the types of bifurcations or singularities that can be present in a gradient-like landscape that, in mathematical terms, can be expressed as a family of potential functions. While CT seemingly only pertains to local bifurcations, used together with other ideas from Dynamical Systems Theory, CT and its key results can be used to gain understanding of the global structure of landscape bifurcations.

Vulval development is an excellent but challenging problem to test this approach. It involves a broadly studied stage in *C. elegans* development, and although there is an extensive amount of data available, it is far from completely understood. The vulva is an adult structure that develops during the larval stages of the hermaphrodite worm. The mature vulva contains 22 cells of different types, and more than 40 genes are involved in its development. It is derived

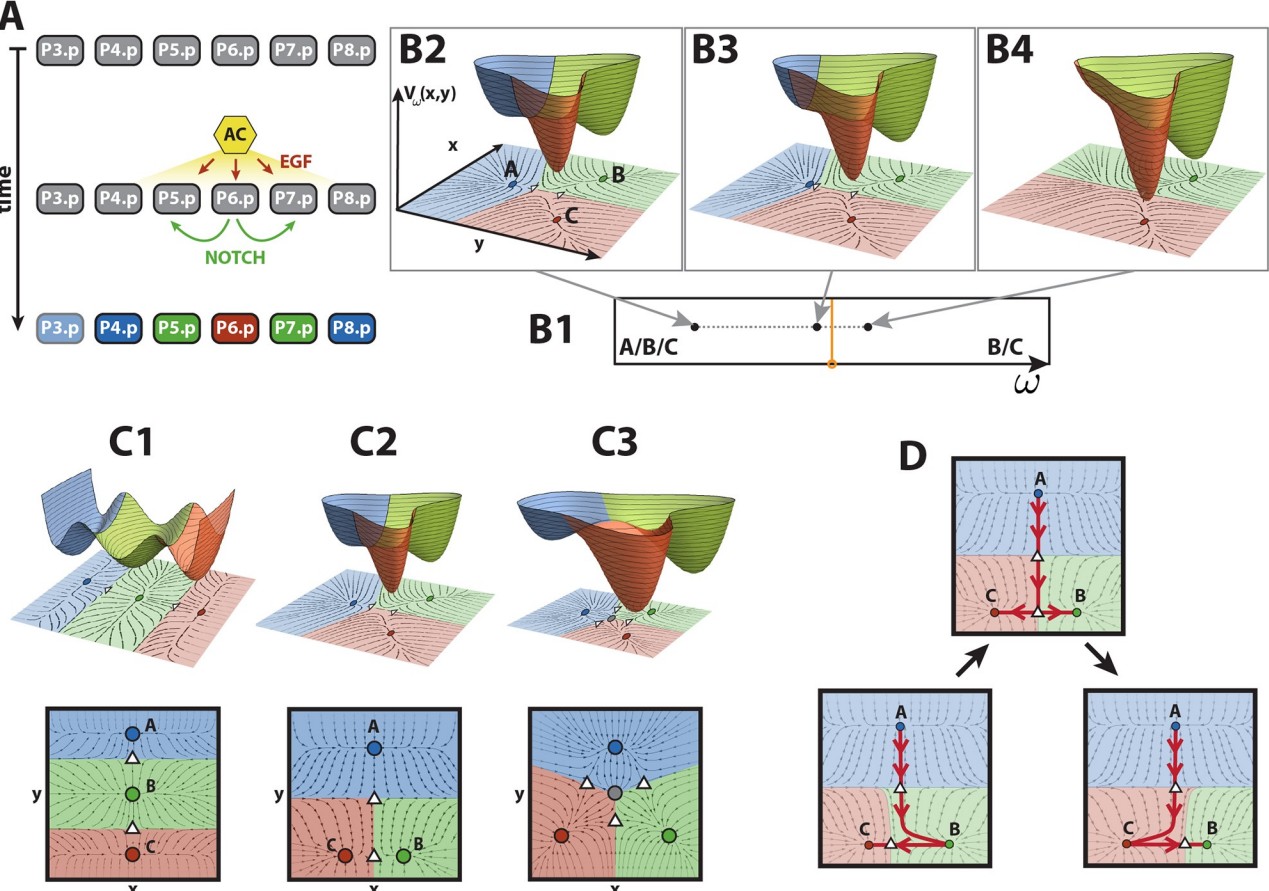

**Fig 1. Vulval development in *C. elegans* and landscape models.** (A) Schematic representation of vulval development in *C. elegans*. Anchor cell (AC) induces vulval precursor cells (VPCs, P3–8.p) to differentiate into three different cell types: primary fate (red), secondary fate (green) tertiary fate (blue). The pattern is controlled by two signals, EGF from the AC (red arrows) and paracrine Notch (green arrows). VPCs are colored according to the WT pattern. P3.p is colored in a shaded blue because it is not included in our study. (B) Example of a bifurcation controlled by a parameter $\omega$ in a landscape with three attractors plotted over the corresponding 2-dimensional flow (B2–4) and its corresponding fate map (B1). Attractors and saddles are represented as circles and white triangles, respectively, on the 2-dimensional flow. Attractors and their corresponding basins of attraction are colored accordingly. (B1) Fate map defined by the parameter $\omega$, where the bifurcation set is colored in orange. (B2) For this value of $\omega$, the landscape contains three attractors $A$, $B$ and $C$. (B3) A change in the parameter of the potential that is now closer to the bifurcation set produces a change in the stability of the landscape, making the valley corresponding to the blue attractor shallower as the attractor approaches the saddle. (B4) The parameter value has crossed the bifurcation set and is now positioned in another region of the fate map. The bifurcation has happened: the blue attractor has bifurcated away and is no longer present in the landscape. (C) The three possible landscape topologies for a process involving three fates, ordered by simplicity. Basins of attraction and attractors are colored according to the fate they could represent. Saddles and repellers are represented as white triangles and gray circles, respectively. (C1) Thom's butterfly catastrophe where the three attractors representing the three different cell types are positioned on a curve. This topology corresponds to the morphogen model and is inconsistent with AC ablation data. (C2) 2-dimensional landscape with two decision points: vulval vs non-vulval fate (top saddle) and primary vs secondary fate (bottom saddle). This is the binary flip with cusp landscape and is our choice of landscape. (C3) Symmetric 2-dimensional landscape where all transitions are allowed. This is the landscape studied in [3, 5]. (D) Schematics of the transitions allowed in the binary flip with cusp model. The unstable manifold of the top saddle can flip from attractor $B$ (bottom left) to attractor $C$ (bottom right) by going through an intermediate state where the unstable manifold of the top saddle directs the flow to the bottom saddle (middle top). This intermediate state is called a heteroclinic connection.

from six ectodermal cells (P3–8.p), called **vulval precursor cells** (VPCs); they are partially differentiated and situated in a row along the antero-posterior axis on the ventral side of the larva, and they all have equivalent developmental potential at the start of the process (Fig 1A).

During vulval development, the VPCs can develop into three different fates: primary (1˚), secondary (2˚) and tertiary (3˚). The first two fates correspond to vulval cells, while the tertiary fate is non-vulval, and cells adopting the tertiary fate fuse with the large syncytial epidermis

hyp7. In the wild type (WT) (i.e. the typical form of *C. elegans* as it occurs in nature), P6.p assumes a primary fate, P5.p and P7.p assume a secondary fate and the rest of VPCs adopt a tertiary fate (Fig 1A). This means that only three precursor cells (P5.p, P6.p and P7.p) form the vulva, while the remainder fuse with the syncytial epidermis [7–9]. P3.p often does not divide, fuses with the hypodermis, and assumes a tertiary fate [3, 7, 10].

This process is orchestrated by the anchor cell (AC), positioned in the gonad of the larva, and VPCs determine their fates through the activation of two signaling pathways [11]: the EGF and Notch pathways. The EGF ligand (*lin-3*) is a signaling molecule secreted by the AC; whereas the Notch ligands, which bind to Notch receptors (*lin-12*), are produced by the VPCs themselves (Fig 1A). The period of competence for the VPCs to respond to these external signals begins in the L2 larval stage and stops shortly after the first round of cell division in L3.

The sequence of events that promote this precise pattern of cells fates (in which cells P4–8.p adopt the fates 3˚2˚1˚2˚3˚), which happens with an accuracy higher than 97% [12], is still unknown. Two models have been proposed to describe the patterning mechanism: the **morphogen model** and the **sequential model**. In the morphogen model, EGF acts as a morphogen and its levels determine the fate of each cell. Consequently, under this model, the fate of a cell depends on the distance to the AC [10]. On the other hand, the sequential model hypothesizes that the anchor cell induces the primary fate in the closest VPC which, in turn, induces the neighboring cells towards secondary fate through activation of the Notch pathway. Cells in which neither of these pathways are activated adopt a tertiary fate [13]. However, there is experimental evidence supporting characteristics of both of these two mechanisms and a more accurate description likely includes aspects of both of these models.

Phenotypes have been determined for a number of mutant worms with alterations in either Notch or EGF signaling. These experiments shed light on the mechanisms that regulate the development of the vulva. For example, it has been observed that a significant reduction of the EGF signal (either by AC ablation at a very early time or by EGF loss of function) produces the pattern 3˚3˚3˚3˚3˚ [14], suggesting that the activation of the EGF pathway is necessary for the induction of the VPCs.

In their pioneering work, [3, 5] use a relatively complex model to quantitatively fit a substantial and complex set of experimental results, and use this to provide a number of quantitative predictions. Building on their work, here we focus on vulval development in *C. elegans* to illustrate a new mathematical framework to develop and quantitatively fit landscape models to biological data, based on CT and approximate Bayesian computation (ABC) fitting.

Given a few reasonable assumptions, we first argue that CT allows us to rationally identify the candidate landscapes. The elementary catastrophes enumerated by CT are polynomial equations in at most two variables and a few parameters, which simplifies the analysis. The central idea is that we can use these relatively simple dynamical systems as modules to build up a global system with the desired characteristics, which dynamics might not be reproducible by a single polynomial equation. Once this model is developed, we show a novel application of ABC methods and demonstrate how coupling CT with ABC fitting to efficiently explore the parameter space provides a useful methodology to fit landscape models to a large amount of data. While the symmetric model developed in [3, 5] assumed that all cell states are equivalent and cells can transition to any fate at all times, with this approach we show that a simpler model where the the non-vulval fate is distinct from vulval ones and to which cells cannot go back to once they are partially differentiated, is also consistent with the data. Taken together, here we show that we can build a fundamentally different and considerably simpler model to the ones in [3, 5] that explains the large amount of data available, and makes a number of interesting predictions that suggest novel ways of understanding biological effects and new experiments to validate them.

## Methods

### A three-way decision landscape

Dynamical Systems Theory allows us to frame the Waddington landscape idea in mathematical terms. The state of a cell at a particular time $t$ will be represented by the position on the landscape, $\boldsymbol{x}(t) = (x(t), y(t))$. The evolution of $\boldsymbol{x}(t)$ in time, representing the differentiation of a cell, will be related to the gradient or steepness of the landscape represented by a potential function $V(\boldsymbol{x})$, and will move downhill in this landscape until it reaches a local minimum (an attractor). Each attractor of $V(\boldsymbol{x})$ corresponds to a stable fate that a cell can adopt. Given an attractor $\boldsymbol{x}^*$, its basin of attraction is comprised of all the points with a trajectory that ends in that given attractor. Finally, the lowest point of the barriers that separate these basins are saddle points of the potential function (Fig 1B).

Now, in order for the trajectory to move from one attractor to another one, the barrier between them needs to disappear. The induction of a new fate is biologically controlled by cues, known as morphogens or signals, that the cell receives and produce a change in its gene expression, pushing the cell towards a new differentiated state. Translating this idea to mathematical terms, the shape of the landscape must be controlled by these signals and its shape will change as these signals change in time. Therefore $V(\boldsymbol{x})$ also depends on some parameters $\omega$, which will relate to the biological signals, becoming the family of functions $V(\boldsymbol{x}, \omega) = V_\omega(\boldsymbol{x})$.

The most important changes in the landscape are the ones in which an attractor is created or destroyed as the parameters change, and these are called bifurcations. These normally happen when an attractor collides with a nearby saddle. An example of this is shown in Fig 1B. From left to right, by changing the parameter $\omega$ of the landscape, the basin of attraction corresponding to the blue attractor, $A$, gradually becomes flatter as the attractor approaches the saddle point. These two points eventually merge together and disappear. This bifurcation would shift the trajectory of a cell starting in the blue attractor towards either the red or green one. The values of the parameters at which bifurcations occur form the **bifurcation set** of the potential function $V$. The bifurcation set of a potential defines regions in the parameter set that correspond to different stability configurations in the landscape, or fate map (Fig 1B).

**Choice of landscape.**   The problem now is to find the simplest possible parameterized family of landscapes that can quantitatively reproduce the substantial set of experimental results summarized in Table 1. A good starting point is to note that some of the landscapes in this family must contain three attractors as, during the process of *C. elegans* vulval development, a VPC must be able to adopt one of the three fates introduced above: primary, secondary and tertiary. If we demand simplicity by not permitting any repelling points in the landscape, there are essentially only two landscapes with three attractors (Fig 1C1 and 1C2). An example of a more complex landscape with a repeller is shown in Fig 1C3, other, even more complex examples, also exist.

As discussed above, cell state transitions occur because changes in the signals a cell receives destabilize the attractor where the cell state lies allowing it to transition to a new one. In considering these landscapes it is important to keep in mind that since the attractors correspond to biological cell states, there must be a clear correspondence between each attractor and a particular state, and this relationship must be tracked as the attractors move due to signal changes. We label them here $A$, $B$ and $C$. The first landscape (Fig 1C1), which is related to Thom's butterfly catastrophe [23], differs from the second (Fig 1C2) in that, with regard to decision-making, it is more restrained. It is characterized by the fact that, in the parameter region where there are three attractors, the identity of the central state $B$ cannot change. The only transitions allowed are $A \rightarrow B$ (by bifurcating $A$ with the top saddle), $B \rightarrow A$ (by bifurcating $B$ with the top saddle), $B \rightarrow C$ (by bifurcating $B$ with the bottom saddle) and $C \rightarrow B$ (by bifurcating $B$ with the

**Table 1. Table of experimental data obtained from the literature [5].**

|  | Experiment | VPC fates (% 1˚, 2˚, 3˚) | | | TD | VD | References |
|---|---|---|---|---|---|---|---|
|  |  | **P4.p** | **P5.p** | **P6.p** |  |  |  |
|  | **Fully penetrant phenotypes** |  |  |  |  |  |  |
| (1) | Wild type | 0,0,100 | 0,100,0 | 100,0,0 | × |  |  |
| (2) | *let-23* mosaic (No EGF receptors in P5/7.p) |  | Wild type |  | × |  | [13, 15] |
| (3) | Half dose of *lin-3* (Half EGF ligand) |  | Wild type |  | × |  | [16] |
| (4) | Half dose of *lin-12* (Half Notch receptor) |  | Wild type |  | × |  | [17] |
| (5) | Notch null, and 2 × WT EGF (2 ACs) | 0,0,100 | 100,0,0 | 100,0,0 | × |  | [17] |
| (6) | No Notch signaling, WT EGF | 0,0,100 | 0,0,100 | 100,0,0 | × |  | [14, 18, 19] |
|  | **Excess EGF** |  |  |  |  |  |  |
| (7) | JU1100 (overexpression of EGF ligand, level to fit) | 18,46,36 | 45.5,54.5,0 | 96,4,0 | × |  | [20] |
| (8) | JU1107 (2.75 × WT EGF) | 2,15,83 | 19,81,0 | 100,0,0 |  | × | [21] |
|  | **Reduced Notch** |  |  |  |  |  | [21] |
| (9) | JU2039 (WT EGF, reduced Notch) | 0,0,100 | 1,89,10 | 100,0,0 |  | × |  |
| (10) | JU2113 (1.25 × WT EGF, reduced Notch) | 4,6,90 | 18,70,12 | 100,0,0 |  | × |  |
|  | **Excess EGF & ectopic Notch** |  |  |  |  |  | [21] |
| (11) | JU2091 (1.25 × WT EGF, ectopic Notch) | 0,0,100 | 0,100,0 | 100,0,0 |  | × |  |
| (12) | JU2089 (1.79 × WT EGF, ectopic Notch) | 0,6,94 | 1,99,0 | 100,0,0 |  | × |  |
| (13) | JU2092 (2.75 × WT EGF, ectopic Notch) | 5,24,71 | 0,100,0 | 100,0,0 |  | × |  |
|  | **Reduced EGF** |  |  |  |  |  | [21] |
| (14) | CB1417 (*lin-3*(e1417) EGF hypomorph) | 0,0,100 | 1,99,0 | 54,0,46 |  | × |  |
| (15) | JU2095 (mild ectopic Notch × EGF hypomorph) | 0,1,99 | 0,15,85 | 72,0,28 |  | × |  |
|  | **Phenotypes following anchor cell ablation** |  |  |  |  |  | [22] |
| (16) | L2 lethargus | - | 0,0,100 | 0,0,100 | × |  |  |
| (17) | Early L3 | - | 1.5,21,77.5 | 18,18,64 |  | × |  |
| (18) | DU divided | - | 0,54.63,45.37 | 31,38,31 | × |  |  |
| (19) | VU divided | - | 4,90,6 | 52,48,0 |  | × |  |
| (20) | 3˚ divided | - | 1,99,0 | 65,35,0 |  | × |  |
| (21) | 2-cell stage | - | 1,99,0 | 93,7,0 |  | × |  |

Probabilities have been converted into percentages for clarity. Fates for P4.p and P8.p (or P5.p and P7.p) have been averaged, since we assume that the pattern is symmetrical around the AC, and P3.p is not considered in this study as it often does not divide and fuses with the hypodermis, assuming tertiary fate. Abbreviations: TD = training data; VD = validation data. Developmental stages: (L2 lethargus) lethargic L2; (Early L3) early L3; (DU divided) Dorsal Uterine precursor cells dividing or divided once; (VU divided) Ventral Uterine precursor cells dividing or divided once; (3˚ divided) 3˚ cells have divided; (2-cell stage) all Pn.p cells have divided once.

bottom saddle). This means that cells need to pass through state *B* in order to go from *A* to *C*. This landscape, in fact, corresponds to the morphogen model, in which a VPC would stay in tertiary fate (*A*) under a low EGF signal, would transition from tertiary fate to secondary fate ($A \rightarrow B$) in response to a middle EGF signal, and to primary fate ($A \rightarrow B \rightarrow C$) in response to a high EGF signal. This landscape, however, is not consistent with the AC ablation experiments in [22], where early AC ablation experiments show an equal escape of cells to secondary and primary fates (Table 1) at early AC ablation times, since this means that cells take the same time to go from *A* to *B* than from *A* to *C*.

On the other hand, for the alternative landscape (Fig 1C2), the fact that the unstable manifold of the top saddle *A* can flip from *B* to *C* via a heteroclinic connection allows extra transitions and for either of *B* and *C* to be the central attractor (Fig 1D). Therefore, in this case, the transition $A \rightarrow C$ is also allowed and cells can go to the state *C* without passing through *B*,

which is consistent with the AC ablation experiments mentioned earlier. We call the landscape in Fig 1C2 the **binary flip with cusp landscape**.

The more complex landscape containing a repeller shown in Fig 1C3 is essentially that used in [3, 5], where it was shown that it can reproduce the data we are concerned with. In this model, all attractors are equivalent to each other and all bifurcations are allowed. Biologically, this means that all cell states are equivalent and cells can transition to any fate at all times.

Our aim is to show that the same data set can also be explained by the considerably simpler binary flip with cusp landscape. This results in a different biological interpretation of the decision-making process. In this picture, not all the fates are equivalent, but the tertiary fate is a special fate. It is the default state for a cell if it does not receive any signal; and cells do not return to it once partially induced, even if the signals are switched off (as AC ablation experiments show in Table 1 [22]). Transitions from fate 3˚ to fate 1˚, and fate 3˚ to fate 2˚, are allowed [24]. Also, the transition from fate 2˚ to fate 1˚ is achieved by increasing the EGF signal and turning Notch signal off [17]. These reasons lead us to hypothesize that the topology of decisions is represented by that of the binary flip with cusp landscape. First, cells will decide whether to differentiate into a vulval fate or stay non-vulval (determined by a bifurcation between the blue attractor *A* and the top saddle in Fig 1C2) and, if they do so and surpass the top saddle, then they will decide whether to become 1˚ or 2˚ fated cells (determined by bifurcations between the bottom critical points in Fig 1C2).

**Building the landscape.** In order to find a representative model for this landscape, we could attempt to use CT to search for a parameterized set of functions such that the parameterized landscape is given by the gradient flows of these functions. However, this approach is not practicable for very complex landscapes. Therefore we employ an alternative method that can be used in this more general context. We use catastrophe and bifurcation theory to analyze the local structure and then use ideas from differential topology to glue the local models together. In our case this involves composing two simple catastrophes: a fold (commonly also known as a saddle-node bifurcation) that will explain the non-vulval vs vulval bifurcation (bifurcation between attractor *A* and top saddle in Fig 1C2), and a cusp catastrophe that will explain the primary vs secondary fate bifurcation (bifurcation between attractors *B* and *C* and bottom saddle in Fig 1C2). For gradient and gradient-like systems these are the only local bifurcations that occur generically and are universal in one and two-parameter systems [25].

The fold or saddle-node is the only bifurcation that appears generically in one-parameter gradient-like systems of any dimension $n$, and it has the normal form $\dot{x}_1 = f_{\text{fold}}(x_1, c) = -x_1^2 - c, \dot{x}_i = -\lambda_i^2 x_i$ ($i = 2, \ldots, n$), where $(x_1, x_2, \ldots, x_n)$ is a choice of coordinate system, and $c, \lambda_i \in \mathbb{R}$ are parameters (Fig 2A). Any system displaying a saddle-node bifurcation can be transformed into this [25]. Here, when the parameter $c < 0$, the system has two equilibria, an attractor and a saddle, and when $c > 0$ it has none (see S1 Appendix for more details). Thus, it can represent the disappearance of the tertiary fate upon receiving signals. The distance between these two equilibria grows like $\sqrt{c}$. We will use it to define the attractors corresponding to 3˚ fate and the point of transition between this non-vulval state and the other two vulval states (bifurcation between 3˚ attractor and top saddle in Fig 2C).

For 2-parameter gradient-like systems the only other generic bifurcation is the cusp. Similarly, the relevant normal form is $\dot{x}_1 = f_{\text{cusp}}(x_1, a, b) = -4x_1^3 - 2ax_1 - b, \dot{x}_i = -\lambda_i^2 x_i$ ($i = 2, \ldots, n$), where $(x_1, x_2, \ldots, x_n)$ is a choice of coordinate system, and $a, b, \lambda_i \in \mathbb{R}$ are parameters. In this case, when the discriminant $\Delta = 8a^3 + 27b^2 < 0$, the normal form system contains two minima and a maximum; when $\Delta = 0$, it contains one minimum and a degenerate point; and if $\Delta > 0$, it contains just one attractor (see Fig 2B and S1 Appendix for more details). We will use

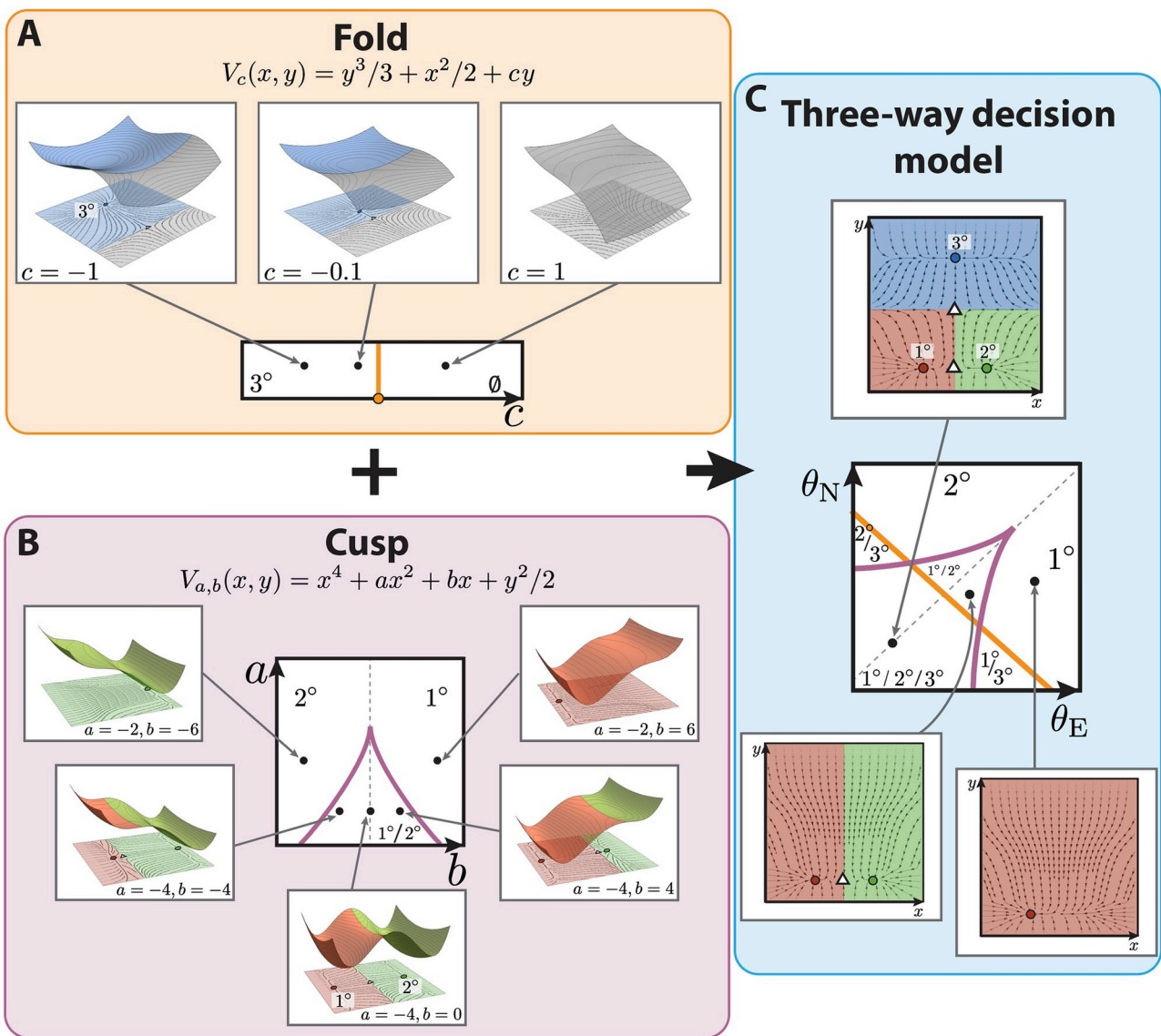

**Fig 2. Binary flip with cusp model of vulval development in *C. elegans*.** (A) The fold catastrophe, controlled by the parameter *c*, defines the stability of the attractor representing 3˚ fate. The fate map is represented at the bottom, showing that the fold has a bifurcation at $c = 0$ (in orange), separating two stability regions: one containing 3˚ fate (for $c < 0$) and one where the fate disappears ($c > 0$). At the top, from left to right, different examples of landscapes for different values of *c*. For $c < 0$, the potential has two critical points in the state space: a minimum (blue circle) and a maximum (white triangle), such as the one represented on the top panels. Close to the bifurcation point, $c = 0$, the two critical points get near each other to merge together. For values of $c > 0$, the potential does not contain any critical point. (B) The cusp catastrophe, controlled by the parameters *a*, *b*, defines the stability of the attractors representing 1˚ and 2˚ fates. The fate map, at the center, shows that the cusp has a bifurcation set determined by $\Delta = 8a^3 + 27b^2 = 0$ (purple cusp), separating three stability regions: one containing 1˚ and 2˚ fate (for $\Delta < 0$) and two where only one of the two fates is present ($\Delta > 0$). Examples of landscapes determined by different values of *a*, *b* are shown; in each case, 1˚ and 2˚ are represented by red and green circles, respectively, and the saddle between them is represented by a white triangle. On the cusp mid-line (gray dotted line in the fate map) the two fates are equally probable. Getting closer to the cusp favors one fate over the other, by the saddle approaching one of the attractors. Outside the cusp, the landscape only has one attractor, either 1˚ or 2˚. (C) The two catastrophes are combined to generate a 2-dimensional flow with two saddles (white triangles) and three attractors representing each fate (blue, green and red circles represent 3˚, 2˚ and 1˚ fates, respectively). By mapping the signals $\theta_E$, $\theta_N$ onto the control parameters *a*, *b*, *c*, a fate map is obtained, defining the available fates in the landscape for different signaling profiles. The presence of 3˚ fate is controlled by the fold line, as shown in (A), and the stability of the 1˚ and 2˚ fates is controlled by the position of the signaling profile with respect to the cusp line, as shown in (B).

it to define the attractors corresponding to fates 1˚ and 2˚ and the saddle between them (Fig 2B and 2C).

We merge the fold and the cusp catastrophe models into a two-dimensional dynamical system (Fig 2C), the mathematical details of which are in Box 1 and S1 Appendix.

The steady states of this system will lie either on the *x*-axis, in which case their *x*-coordinates will be given by the steady states of the cusp (i.e. there will be three, two or one steady states on the *x*-axis); or they will have *y*-coordinates which are the zeros of a fold, (there will be two, one or none depending on *c*) in which case their *x*-coordinates will be 0 (see Box 1, Fig 2C and S1 Appendix for more details).

We choose the attractor with zero *x*-coordinate to represent the 3˚ fate, the attractor with zero *y*-coordinate and negative *x*-coordinate to represent the 1˚ fate and the attractor with zero *y*-coordinate and positive *x*-coordinate to represent 2˚, as shown in Fig 2C. Taken together, parameter *c* will control whether 3˚ fate is present in the landscape, and *a*, *b* will control whether the attractors representing 1˚ and 2˚ fates will be present in the landscape.

The advantage of building the landscape from these two catastrophes is that, firstly, mathematically one has the universality property mentioned above and further described in S1 Appendix and, secondly, that the position and stability of the various restpoints and their dependence on the parameters is transparent and, as shown in Box 1, one has complete understanding of the regions of the parameter space with common stability, being able to characterize the shape of the landscape for any parameter value.

**Dependence upon the morphogens.** The vulval development process is controlled by induction of the VPCs by the anchor cell via the EGF signal and by lateral signaling through Notch. Therefore, to connect our model to the biological data we need to define how the control parameters *a*, *b* and *c* defined previously depend upon the levels $\theta_E$ and $\theta_N$ of the EGF and Notch signals, respectively.

As we describe in Box 2, in more detail, we choose a flexible linear dependence of the control parameters upon the biological signals.

Assuming that all VPCs are equivalent at the beginning and that they start at the basin of attraction of the default state, 3˚ fate, the transformation can be constrained by noting the following facts:

1. The origin of the $(\theta_E, \theta_N)$ coordinate system is assumed to correspond to a point in which there is tristability in the landscape, as once induced all three fates are stable after the removal of signals (see Table 1) [7].

2. The signals EGF and Notch drive the system out of the tristable region.

3. The Notch null, 2 × WT EGF perturbation results in an induction of P5.p and P6.p cells to adopt primary fate (see Table 1), and therefore this signaling profile lies outside the tristable region.

4. Similarly for the Notch null with WT EGF perturbation, P6.p cell adopts primary fate and therefore its signaling profile lies outside the tristable region.

More details can be found in S1 Appendix.

Next, to complete our model, we need to define the signaling profile of each cell in time, which we elaborate in Box 3. As we lack dynamic signaling data, such as could be obtained from live-cell reporters, we make the following assumptions. First, to model the EGF signal, $\theta_E^i(t)$, received by a cell P*i*.p, we consider that the EGF ligand is secreted by the AC, and therefore the signal received by a cell will depend on its proximity to the AC. We assume this signal is not instantaneous, so its increase is modeled as a monotonously increasing function in time.

### Box 1: Mathematical details of the model

We merge the fold and the cusp catastrophes into the following model, which will be used to model the evolution of the state of a single VPC in time, represented by $x(t) = (x(t), y(t))$:

$$\begin{cases} \dot{x} & = & H(-y)f_{\mathrm{cusp}}(x, a, b) - (1 - H(-y))x = f_1(x, y, a, b, c, M) \\ \dot{y} & = & yf_{\mathrm{fold}}(y - M, c) = f_2(x, y, a, b, c, M), \end{cases} \tag{1}$$

where $H$ is a Heaviside function that is equal to 0, if $y$ is less than 0, and equal to 1 otherwise. Strictly speaking to get a smooth dynamical system we should smooth the Heaviside function but, as will become clear and is shown in S1 Appendix, this does not affect the analysis shown here.

The attractors of this system are obtained by solving $f_1 = f_2 = 0$. Since $f_2$ does not depend on $x$, one can quickly solve it to obtain the $y$-coordinates of the critical points, which are given by $y_1^* = M + \sqrt{-c}$, $y_2^* = M - \sqrt{-c}$ and $y_3^* = 0$. If $f_1 = 0$ is now solved with each of the possible values of $y$ obtained, the $x$-coordinate of the critical points with $y = y_1^*$ or $y = y_2^*$ is equal to 0; while the $x$-coordinates of the critical points with $y = y_3^* = 0$ are given by the zeros of the cusp, i.e. $f_{\mathrm{cusp}}(x^*, a, b) = 0$. The parameter $M$ controls the relative position of top and bottom saddles in Fig 2C.

The stability of these points can be obtained by looking at the Jacobian matrix of the system. It can be shown that the point $(0, y_1^* = M + \sqrt{-c})$ is stable (blue attractor in Fig 2C, assigned to represent 3° fate), the point $(0, y_2^* = M - \sqrt{-c})$ is unstable (white top triangle in Fig 2C), while the points laying on the $x$-axis can be either two stable points (representing 1° and 2° fates) separated by a saddle (as in Fig 2C), a stable point and a degenerate point or just a stable point. We refer the reader to S1 Appendix for more details and for more examples of landscape configurations.

Taken together, the parameter $c$ controls the existence and positions on the $y$-axis of critical points with positive $y$ coordinate (which can be two, one or none), while the parameters $a$, $b$ control the the positions on the $x$-axis and existence of critical points with zero $y$-coordinate (which can be three, two or one). With this model, 3° attractor can be bifurcated away and be removed from the landscape, however, either 1° or 2° fates need to be present and both cannot be bifurcated away to give a landscape where only 3° fate is present, which is intrinsically different from the model in [3, 5].

Since the 3° attractor is controlled by the fold we know that if $c < 0$, the attractor corresponding to this fate will be present, no matter the values of $a$ and $b$. On the other hand, the presence of the attractors corresponding to 1° and 2° fates depends only on the values of $a$ and $b$, specifically on the value of the discriminant $\Delta = 8a^3 + 27b^2$ as explained above, independently from the value of the parameter $c$, and therefore their presence will be controlled by the value of $\Delta$.

One can then compute the bifurcation set of the landscape in Eq 1, and obtain:

$$\mathcal{B} = \mathcal{B}_1 \cup \mathcal{B}_2 \equiv \{(a, b, 0) : a, b \in \mathbb{R}\} \cup \{(a, b, c) \in \mathbb{R}^3 : 8a^3 + 27b^2 = 0\}, \tag{2}$$

where $\mathcal{B}_1$ and $\mathcal{B}_2$ are represented in orange and purple colors, respectively, in Fig 2. Bifurcations will happen at and only at the values of the parameters in $\mathcal{B}$. This bifurcation set divides the control space into regions with common stability, some of which are represented in S1 Appendix. More details can be found in S1 Appendix.

## Box 2: Mathematical details of the map between control parameters and signals

We will postulate a flexible functional form for the relationship between the control parameters $a$, $b$ and $c$ and the signaling levels $\theta_E$ and $\theta_N$, and then fit this to the data. We assume that the parameters $a$, $b$ and $c$ are affine functions of $\theta_E$ and $\theta_N$ so that

$$
\begin{pmatrix} a \\ b \\ c \end{pmatrix} = \begin{pmatrix} m_{11} & m_{12} \\ m_{21} & m_{22} \\ m_{31} & m_{32} \end{pmatrix} \begin{pmatrix} s\theta_E \\ l\theta_N \end{pmatrix} + \begin{pmatrix} q_1 \\ q_2 \\ q_3 \end{pmatrix} \tag{3}
$$

where $m_{i,j} < 1$ and $s$, $l$ are scaling parameters of the signals into the control space. Then $(a, b, c)$ lies on the plane $\pi_T$ given by $Aa + Bb + Cc = D$ where $A = m_{31}\, m_{22} - m_{21}\, m_{32}$, $B = m_{11}\, m_{32} - m_{31}\, m_{12}$, $C = m_{12}\, m_{21} - m_{11}\, m_{22}$, and $D = -Aq_1 - Bq_2 - Cq_3$

This plane intersects the bifurcation set $\mathcal{B}$ in different subsets depending on the values of the $m_{ij}$ and the scaling parameters $s$, $l$, while the origin of the $(\theta_E, \theta_N)$ coordinate space in that plane will be determined by the parameters $q_i$. Consequently, the bifurcations can be visualized in the $(\theta_E, \theta_N)$-signal space by intersecting the plane and the bifurcation set $\mathcal{B}$ and this gives a fate map for the cell as a function of the signals EGF and Notch (See Fig 2C, and S1 Appendix for more details).

## Box 3: Mathematical details of the signaling dynamics

Finally, we model the signal profile of each cell in time in the following way. To reduce unnecessary dimensions, we only model the cells P4–6.p as the pattern is generally symmetric around the AC and, as mentioned in the introduction, P3.p often assumes tertiary fate. Following the approach in [3, 5], we assume that the difference of EGF signal between consecutive cells is regulated by a scaling parameter $\gamma \leq 1$, which derivation follows from the diffusion of the EGF ligand. However, we also model the increase of EGF signal in time as a monotonous increasing function $\sigma(t)$:

$$
\theta_E^i(t) \in \{\gamma^2\sigma(t), \gamma\sigma(t), \sigma(t)\}, \tag{4}
$$

where we write $\sigma(t)$ as

$$
\sigma(t) = \frac{1 + \tanh\left(H_E t + M_E\right)}{2}. \tag{5}
$$

Regarding Notch signaling, following [3, 5], we define the production of Notch by a cell by a sigmoidal function $L$, that depends on the current state $\boldsymbol{x}(t)$ of the cell:

$$
L(\boldsymbol{x}(t)) = \frac{1 + \tanh\left(n_0 + \boldsymbol{n}_1 \cdot \boldsymbol{x}(t)\right)}{2}, \tag{6}
$$

where $n_0$ and $\|\boldsymbol{n}_1\|$ are constants and the vector $\boldsymbol{n}_1$ has negative $x$-coordinate so that $L$ increases as the state of the cell approaches the basin of attraction of 1° fate. We now define $\theta_N^i(t)$ of a cell P$i$.p to be proportional to the sum of the autocrine and the paracrine Notch signal received by the cell. Also following [3, 5], the autocrine signal is scaled by a parameter $\alpha > 0$ which parametrizes the relative importance of autocrine vs

paracrine signaling. We also take into account that 1° fated cells downregulate Notch receptor *lin-12* [19]. This downregulation is related to the cell's production of Notch signaling, scaled by a parameter $l_d$, which controls the strength of such downregulation. Therefore the Notch signal received by a cell P$i$.p is modeled as:

$$\theta_N^i(t) = (1 - l_d L(\boldsymbol{x}^i(t))(L(\boldsymbol{x}^{i-1}(t)) + \alpha L(\boldsymbol{x}^i(t)) + L(\boldsymbol{x}^{i+1}(t)))) \qquad (7)$$

This is consistent with data published in [22], where the activity of the EGF pathway was measured using a transcriptional reporter, and showed a monotonal increase in time, as shown in S1 Appendix. Second, to model $\theta_N^i(t)$, the Notch signal received by a cell P$i$.p, it is known that Notch signal production is a consequence of EGF pathway activation, so that the Notch signal is produced by cells receiving high EGF as they adopt the primary fate. Therefore, we assume cells produce Notch signals as they approach the basin of attraction corresponding to the 1° fate. Moreover, the Notch signal received by a cell is a sum of autocrine Notch via diffusible delta ligands plus paracrine Notch from the neighboring cells. Taking this together, we model $\theta_E^i(t)$ and $\theta_N^i(t)$ as described in Box 3 and S1 Appendix, in more detail.

Finally, to account for variability in the outcomes, we add some random white noise in the dynamics of each cell, which is parameterized by a coefficient of diffusion in the phase space.

If we now combine the model in Box 1 together with the mapping from the signals to the control parameters in Box 2 and the signaling dynamics to which a cell is exposed in time, described in Box 3, we obtain the proposed model.

Each cell will start its trajectory in the basin of attraction representing the tertiary fate and will move in its own dynamic flow determined by Eq 1, the shape of which depends, by the relationship in Eq 3, on the signals it is receiving.

After the period of competence, the fate of a cell is defined by the basin of attraction at which it ended up. The mathematical details of how this is done are presented in S1 Appendix. This now allows us to fit the model to the experimental data.

## Parameter estimation: ABC SMC implementation

Now that a parameterized model has been developed, the next step is to find whether there are parameters that allow the model to reproduce the experimental data in Table 1 and to determine the extent of such parameters. Table 1 contains the probabilities of each VPC (P4.p, P5.p, P6.p) acquiring each fate (1°, 2°, 3°) under twenty one experimental conditions (i.e. 171 data points or probabilities in total). From this data set, we will use a subset of nine experimental conditions as training data set for the model to fit, and the remaining ten as validation results (TD and VD in Table 1, respectively).

Let us denote by $p_{e,f,c}$ the experimental probability of cell $c$ becoming fate $f$ under experimental condition $e$, the values of which are given in Table 1. Similarly, $p_{e,f,c}^{sim}(\boldsymbol{\theta})$ represents the simulated probability of cell $c$ becoming fate $f$ under experimental condition $e$ given parameters $\boldsymbol{\theta}$. Taking a Bayesian approach, we treat the parameters of the model as random variables with a probability distribution. Our goal is to estimate the posterior distribution of the parameters that accurately reproduce the training data set $\boldsymbol{X}_0 = (p_{e,f,c})_{e \in TD}$, and check that they also reproduce the validation data set.

If we denote a parameter vector by $\boldsymbol{\theta}$, the posterior distribution satisfies:

$$\pi(\boldsymbol{\theta} \mid \boldsymbol{X}_0) \propto \pi(\boldsymbol{\theta})\mathcal{L}(\boldsymbol{X}_0 \mid \boldsymbol{\theta}), \tag{8}$$

where $\pi(\boldsymbol{\theta})$ is the prior distribution of the parameters $\boldsymbol{\theta}$, $\mathcal{L}(\boldsymbol{X}_0 \mid \boldsymbol{\theta})$ is the likelihood of the data $\boldsymbol{X}_0$ given the parameters $\boldsymbol{\theta}$ and $\pi(\boldsymbol{\theta}|X_0)$ is the posterior distribution of the parameters $\boldsymbol{\theta}$ given the data $\boldsymbol{X}_0$.

Since it is not possible to find an analytical expression for the likelihood of the model we propose, we approximate it using approximate Bayesian computation (ABC). ABC methods, also known as likelihood-free methods, have been developed for inferring the posterior distribution of the parameters when the likelihood function is either too complex or too expensive to compute but observations of the model can be simulated fairly easily. They have been successfully applied to a wide range of intractable likelihood problems over the past twenty years [26] including population genetics [27], pathogen transmission [28], reaction networks models [29, 30] and epidemic modeling [31], among others. ABC methods use a comparison between the experimental and simulated data to measure the goodness of fit, instead of using the likelihood function, allowing for more flexibility.

In particular, here we take advantage of Sequential Monte Carlo ABC (ABC SMC) to explore the parameter space and determine the posterior distribution of the parameters. Various ABC SMC algorithms have been proposed in the literature [27, 32–34], but here we focus on the fairly general ABC SMC algorithm in [34]. The ABC SMC sampler methodology approximates the posterior distribution by sequentially sampling from a sequence of intermediate distributions, or approximate posterior distributions, $\{\pi_t\}_{0 \leq t \leq T}$, that increasingly resemble the posterior distribution given in Eq 8:

$$\{\pi_t\}_{0 \leq t \leq T} = \{\pi(\boldsymbol{\theta} \mid d(\boldsymbol{X}(\boldsymbol{\theta}), \boldsymbol{X}_0) \leq \varepsilon_t)\}_{0 \leq t \leq T}, \tag{9}$$

where $d$ is a metric that compares the experimental and simulated data, and $\{\varepsilon_t\}_{0 \leq t \leq T}$ is a decreasing sequence of distance values. This ABC method is particularly useful in cases where the number of parameters and the data are high dimensional, as it can be easily parallelised. It can also be tuned to efficiently explore the parameter space maintaining areas of high likelihood via the choice of its perturbation kernel. A summary of how this is done is shown in Fig 3.

Our model depends on a total of 22 parameters: 1 time constant that controls the velocity of the trajectories in the landscape, 10 landscape parameters that map the signaling values into the shape of the landscape and 11 signaling parameters that define the signaling profile that each cell is exposed to in time (see S1 Appendix). Both the model's definition and the experimental data constrain the range of the landscape parameters and inform the choice of their priors, for which we choose flat priors on their support. For the rest of parameters we choose fairly non-informative priors that still reflect our knowledge about their ranges. Moreover, taking into account the dimensionality of the parameter space, we choose to approximate the posteriors by sampling $N = 2 \times 10^4$ particles at each step of the algorithm. More details can be found in S1 Appendix.

In order to compare the experimental and simulated data we define a distance $d$ that measures the level of similarity between two data sets. If $\boldsymbol{X}_0^{TD}$ is the subset of data corresponding to experiments in the training data set (see Table 1), we define the distance between $\boldsymbol{X}_0^{TD}$ and the corresponding simulation $\boldsymbol{X}^{TD}(\boldsymbol{\theta})$ as

$$d(\boldsymbol{X}_0^{TD}, \boldsymbol{X}^{TD}(\boldsymbol{\theta})) = \frac{1}{E}\sum_{e \in TD}\sum_{f=1}^{3}\sum_{c=1}^{3}|p_{e,f,c} - p_{e,f,c}^{sim}(\boldsymbol{\theta})| + \frac{1}{E}\sum_{e \in TD}\sum_{c=1}^{3}|p_{e,4,c}^{sim}(\boldsymbol{\theta})|, \tag{10}$$

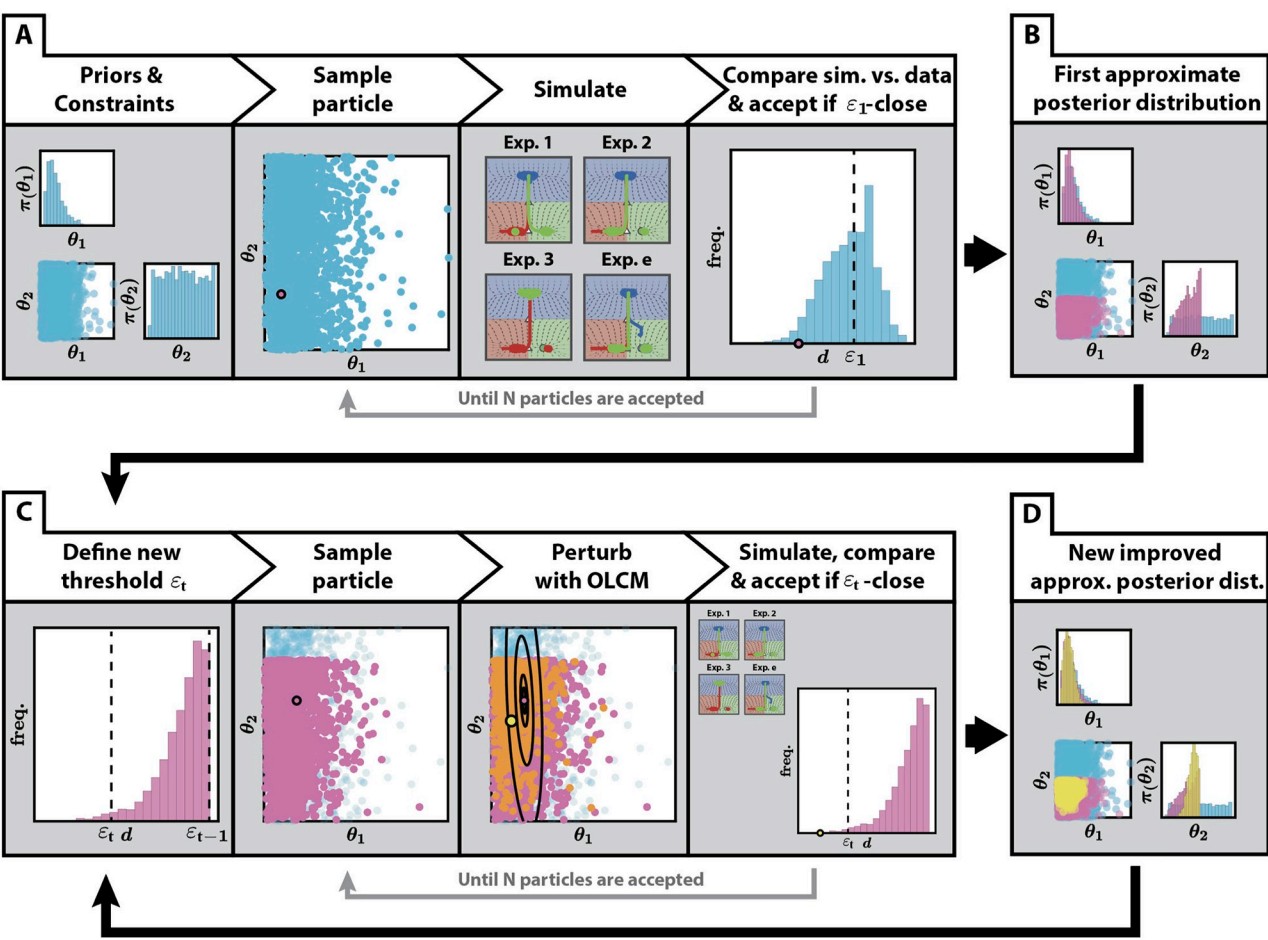

**Fig 3. Application of ABC SMC to the fitting of the binary flip with cusp model.** (A) First step of the algorithm. Having defined priors for the fitted parameters and constraints between them, as well as an initial threshold $\varepsilon_1$, particles are sampled from the priors and accepted if the distance between the simulated data and the experimental data is less than the initial threshold. This is repeated until $N$ particles are accepted. (B) An initial approximation of the posterior distribution is obtained, generated by the $N$ particles accepted in (A). (C) The algorithm then proceeds in a sequential manner. In each step $t$ of the algorithm, a new threshold $\varepsilon_t$ is defined, in our case, the 0.3 quantile of the previous distribution of distances $\varepsilon_{t-1}$. Particles are sampled from the last approximate posterior distribution and perturbed using a Markov Kernel obtained from the Optimal Local Covariance Matrix (OLCM) [35] (black ellipses in the figure represent confidence intervals), (as described in S1 Appendix). This is repeated until $N$ new particles that satisfy the distance threshold are accepted. Also, each new particle is assigned a weight proportional to its prior probability and inversely proportional to the Markov Kernels evaluated at this particle, which control for efficient exploration. (D) After each step, a new improved approximate posterior distribution is obtained, which restricts the values of the parameters to a restricted region of the parameter set.

where $E$ is the number of experiments in the training data set, $p_{e,f,c}$ and $p_{e,f,c}^{sim}(\boldsymbol{\theta})$ are the experimental and simulated probabilities of cell $c$ becoming fate $f$ in experimental condition $e$, respectively, and $p_{e,4,c}^{sim}$ is the proportion of times that our model could not assign a fate to cell $c$ when simulating experiment $e$ (see S1 Appendix for more information). The distance function penalizes parameter values for which our model cannot assign a fate, since we would like to avoid these parameter values. More details about our implementation of the fitting algorithm such as the choice of thresholds sequence and perturbation kernel function can be found in S1 Appendix.

Our choice of experiments for training was based on trying to find a maximally informative set while constraining the number of experiments. For example, we included all experimental conditions (a total of 6) with fully penetrant phenotypes resulting from different signaling

combinations, which explored different regions of the signal space. With regards to partially penetrant phenotypes, we included one experimental condition involving EGF overexpression. We expected that including these experiments would inform the fitting of the values of the scaling parameters $s$ and $l$ and other parameter values related to the strength of the signals and the cross-talk between them. To constrain the temporal features of the model, we also added two AC ablation conditions which we considered the most informative ones, expecting that the model would be able to reproduce the rest. The particular details of the fitting can be found in S1 Appendix.

A description of the methodology, software implementation in MATLAB, and instructions for use are publicly available in the following GitHub repository: https://github.com/ecamacho90/VulvalDevelopment.

## Results

### Fitting the model to the training data: A two-step decision

The results after 14 generations of ABC SMC are shown Figs 4 and 5, S1 and S2 Figs, and S1–S13 Videos, which show excellent agreement overall between the simulations and the data. We are able to reproduce both fully and partially penetrant phenotypes.

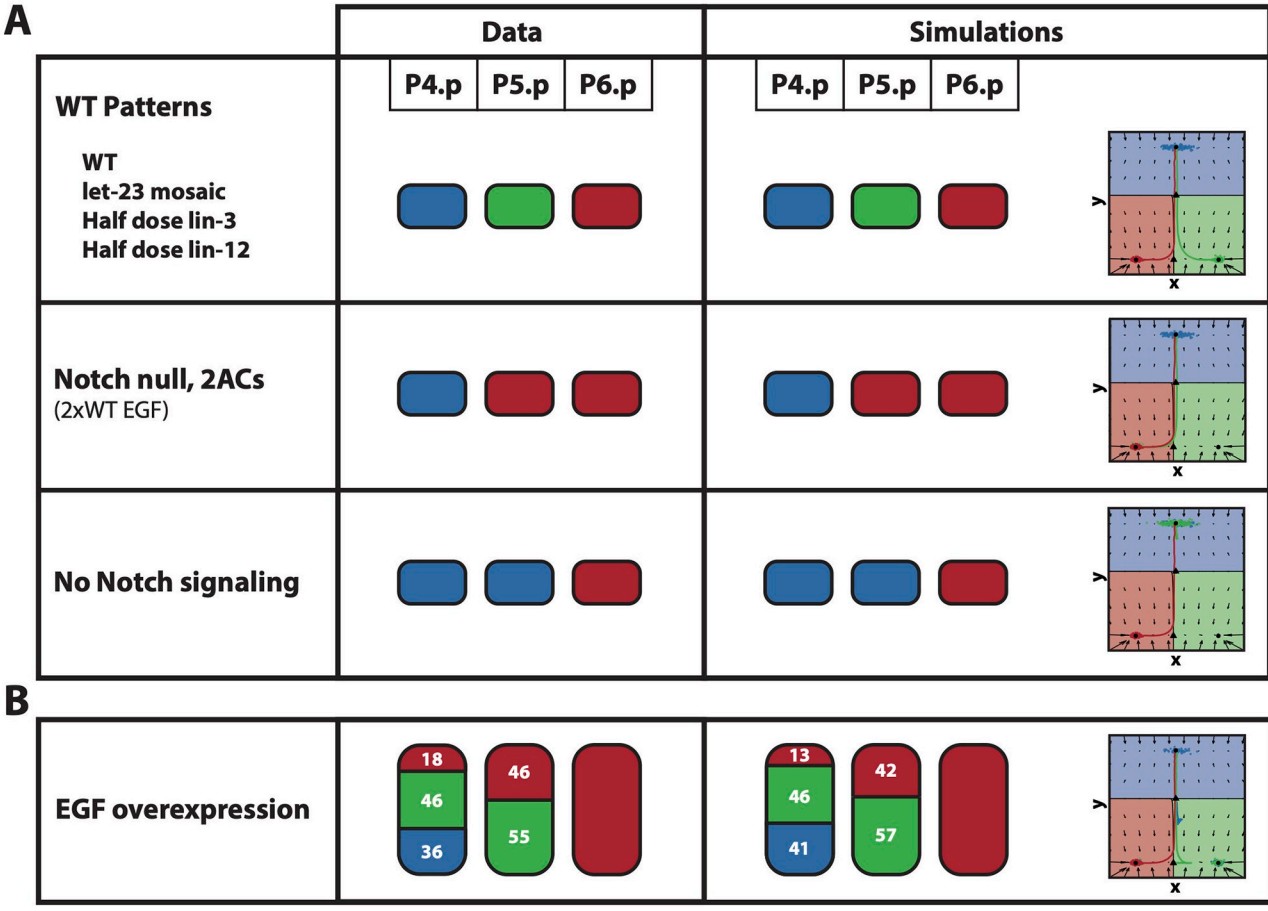

**Fig 4. Fitting of the training data.** (A) The patterns of the 6 fully penetrant phenotype perturbations considered in the training data set are correctly fitted by the binary flip with cusp model. (B) The pattern of the partially penetrant phenotype given by the EGF overexpression perturbation considered in the training data set is also correctly fitted by the model. On the left, the experimental patterns. On the right, the mean simulated patterns and trajectories on the landscape. In both the experimental and simulated patterns, primary, secondary and tertiary fates are represented by the red, green and blue colors, respectively, and proportions higher than 90% have been rounded. On the landscape, the mean simulated trajectories for the particle with best overall approximation for P4.p, P5.p and P6.p are colored in blue, green and red, respectively.

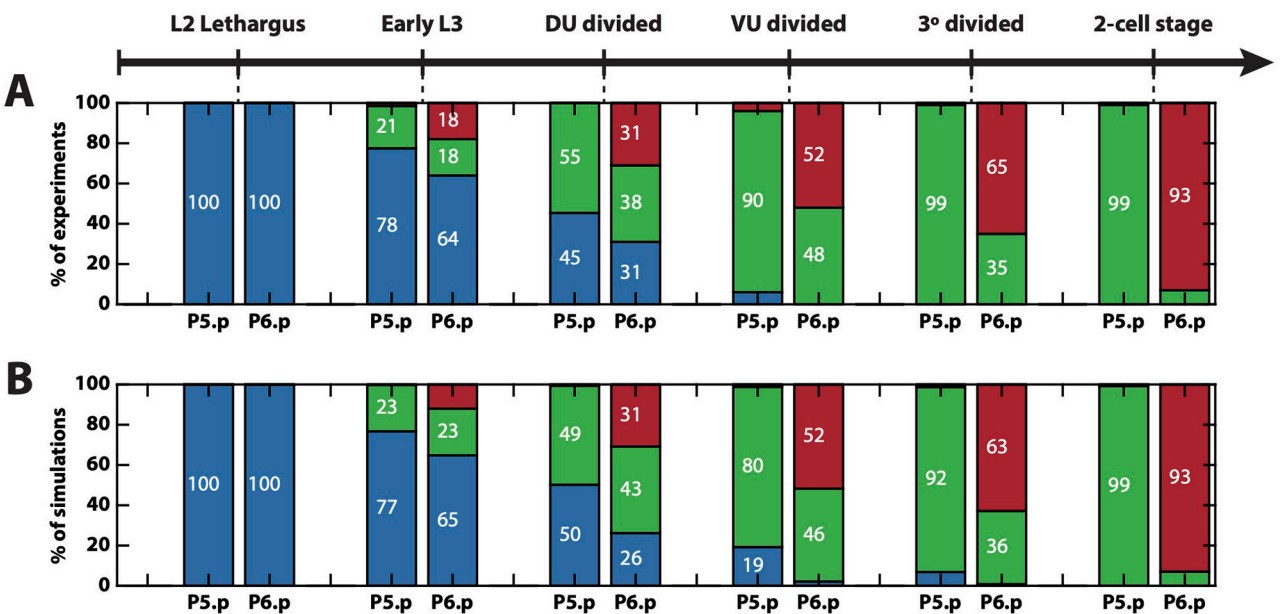

**Fig 5. AC ablation fitting and validation.** (A) Patterns observed when the AC is ablated at different developmental stages. (B) Mean simulated patterns for the different AC ablation conditions. Only proportions for P5.p and P6.p are available, which are presented in blue, green or red representing tertiary, secondary or primary fate, respectively.

S1 Video shows how the simulated VPCs pattern in the WT case. The two-step decision logic of non-vulval vs vulval followed by primary vs secondary fates can be clearly seen. As P6.p starts receiving EGF signal, the cell leaves the attractor corresponding to 3˚ fate, i.e. the first decision is made, and moves into the regions of the fate map and landscape containing vulval fates. The increase in EGF signal and lack of Notch signal from its neighbors, positions its signaling profile in a region of the fate map where only primary fate is stable, and therefore, its trajectory on the landscape moves towards this attractor. A similar effect happens for P5.p but, due to the higher distance to the AC resulting in lower EGF signal, but higher Notch signal from the neighboring P6.p differentiating to primary fate, the cell differentiates towards 2˚ fate. Finally, the signals received by P4.p are not enough for it to escape the attractor corresponding to 3˚ fate, and therefore the cell differentiates into the non-vulval fate. S2–S7 Videos show the differentiation of VPCs in mutants 2–7 in Table 1.

It is worth mentioning that, in our model, due to the lack of information about the exact timings of the stages at which AC ablations were performed, relative to each other, and also about the EGF and Notch signal dynamics, the increase in EGF and Notch signals are modeled by monotonically increasing functions, which are initially hypothesized to be sigmoidal functions of the simulation time. We find that the fitting of the data suggests a slight modification of the EGF dependence on time (see S1 Appendix) which then allows for the model to fit the AC ablation data (Fig 5 and S8–S13 Videos). This is probably due to the fast dynamics around the bifurcation point between the attractor corresponding to tertiary fate and the saddle. More details can be found in S1 Appendix.

Analysis of the evolution of the approximate posterior distributions of the parameters at each step of the ABC algorithm shows that most of the parameters are very well constrained by the data (S1 and S2 Figs), with the exception of the scaling parameters $s$ and $l$ that correlated with the parameter $q_3$ of the linear transformation that determines the position of the fold line. Also, the parameter $l_d$, which controls a direct coupling between EGF and Notch pathways,

was not crucial to reproduce the data, and this is especially important because, as we will show in the following section, the model is able to reproduce epistatic effects without this coupling. Finally, the exponential decays $\lambda_E$ and $\lambda_N$ were only constrained such there was sufficient time to allow the system to return to zero-signal condition (S1 Appendix). Interestingly, the noise magnitude represented by the parameter $\sigma_{dif}$ was strongly constrained, suggesting that the data strongly constrained noise-driven fluctuations. Moreover, the fitting did not show strong parameter correlations with the exception of parameters $H_E$ and $M_E$, defining the sigmoidal increase of EGF signal in time. Also, it is important to note that the parameters $\gamma$ and $\mathbf{n}_1$ that control the balance between the morphogen and sequential models were strongly constrained. In particular, $\gamma = 0.22 \pm 0.06 < 1$ and $\mathbf{n}_1$ pointed towards first fate, so we can infer from the data that both morphogen and sequential features were necessary.

Finally, we also observe that the results of fitting suggest that the geometry of the fate map is strongly constrained by the data, the details of which are shown in S1 Appendix. Taken together, our results suggest that aspects from both the morphogen and the sequential model are important for the correct patterning, which combine into a two-step decision logic determined by the fitted fate map and controlled by the dynamics of the EGF and Notch signals received by the cells.

## Validating the model: The model reproduces epistasis between EGF and Notch

There is an extensive amount of experimental data available with different combinations of signaling perturbations. In particular, [21] provides quantitative signaling data for many perturbation lines. We compiled a set of mutants from the literature and tested whether our model was able to reproduce this data (Table 1). An important feature of the model developed in [3, 5] is that it can reproduce epistatic effects between the signals, i.e. a mutation in EGF signal can alter the effect of Notch and vice versa. Here we show that our model can also explain these effects without including any direct coupling between the pathways.

As shown in [21], an EGF overexpression perturbation of 2.75-fold with respect to the WT (based on measured *lin-3* mRNA levels), named JU1107, increased P5.p induction towards primary fate, and P4.p towards secondary fate. This is an epistatic event, as an increase in EGF signal increases the secondary fate in P4.p, promoted by Notch signaling. Our model correctly reproduces these features as shown in Fig 6A and Table 2. This is achieved because a 2.75-fold increase in the EGF signal of P5.p locates its signaling profile close to the cusp mid-line in the signal space, where primary and secondary fates are equally probable. In turn, primary-fated P5.p cells signal through Notch to their P4.p neighbors which are receiving more EGF than the WT P4.p cells. This positions P4.p closer to the fold line, in a region where secondary and tertiary fates are equally probable (Fig 6A).

Reduced Notch perturbations do not have a strong phenotype unless crossed with an EGF overexpression mutant (see Table 2), in which case the probability of P5.p differentiating into secondary fate is slightly reduced. We checked whether our model was capable of reproducing these features. Since the level of reduction of Notch is not quantified in the experimental settings, we first fitted a multiplicative magnitude of Notch reduction that reproduced the data. With a Notch reduction of $0.4 \times$ WT, our model is able to reproduce both mutant JU2039, where EGF is WT and there is no strong phenotype, and mutant JU2113, where EGF is slightly increased by 1.25-fold and secondary fate in P5.p is destabilized, increasing the probability of primary and tertiary fates. This fitted fold increase is, in fact, similar to the one considered in [5]. The model predicts that, under JU2039 signaling regime, P5.p would stay in a region of the signal space where secondary fate is still predominant, and therefore the pattern would not

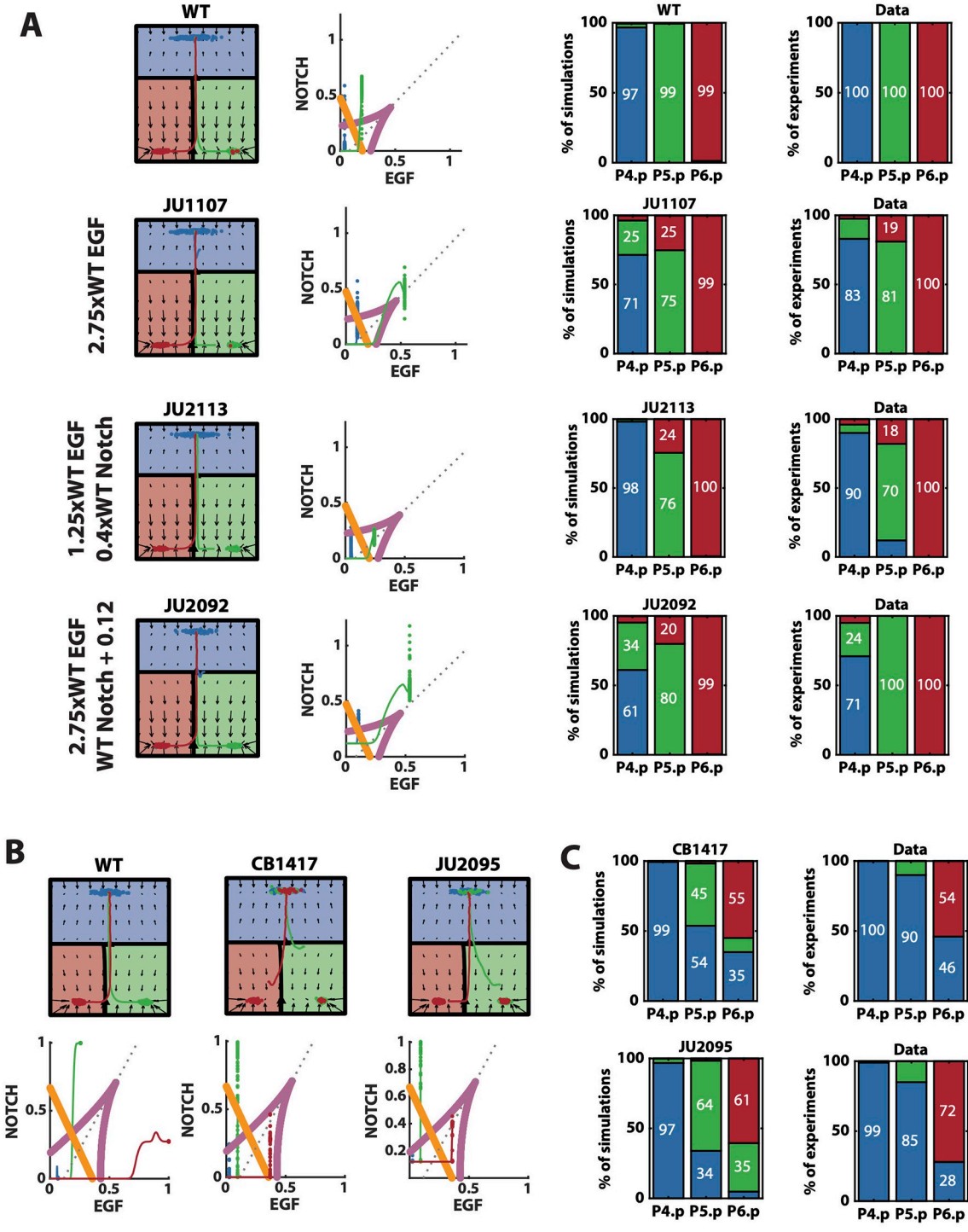

**Fig 6. Model's further validations.** (A) Left: Representative mean trajectories on the landscape and signal space for simulated WT vulva and different signal perturbations included in the validation data set. Blue, green and red trajectories represent the evolution of the state of P4.p, P5.p and P6.p, respectively. Right: Mean simulated and experimental patterns for the different perturbations shown on the left. Blue, green and red represent tertiary, secondary and primary fates, respectively. (B) Representative mean trajectories on the landscape and signal space for simulated WT vulva and EGF hypomorph perturbations. Blue, green and red trajectories represent the evolution of the state of P4.p, P5.p and P6.p, respectively. (C) Mean simulated and experimental patterns for the EGF hypomorph perturbations. Blue, green and red represent tertiary, secondary and primary fates, respectively.

**Table 2. Comparison between experimental and simulated data for the validated experimental conditions.**

| Experiment | VPC fates | | | | | | | | | |
|---|---|---|---|---|---|---|---|---|---|---|
| | | P4.p | | | P5.p | | | P6.p | | |
| | % | 1˚ | 2˚ | 3˚ | 1˚ | 2˚ | 3˚ | 1˚ | 2˚ | 3˚ |
| **Excess EGF** | | | | | | | | | | |
| JU1107 (2.75 × WT EGF) | | 2 | 15 | 83 | 19 | 81 | 0 | 100 | 0 | 0 |
| | | 3±5 | 25±18 | 72±19 | 25±18 | 75±18 | 0±0 | 99±3 | 1±1 | 0±0 |
| **Reduced Notch** | | | | | | | | | | |
| JU2039 (WT EGF, reduced Notch) | | 0 | 0 | 100 | 1 | 89 | 10 | 100 | 0 | 0 |
| | | 0±0 | 0±1 | 99±1 | 16±8 | 80±10 | 4±4 | 99±1 | 1±1 | 0±0 |
| JU2113 (1.25 × WT EGF, reduced Notch) | | 4 | 6 | 90 | 18 | 70 | 12 | 100 | 0 | 0 |
| | | 1±1 | 3±3 | 98±3 | 25±12 | 76±12 | 0±1 | 100±1 | 0±1 | 0±0 |
| **Excess EGF & ectopic Notch** | | | | | | | | | | |
| JU2091 (1.25 × WT EGF, ectopic Notch) | | 0 | 0 | 100 | 0 | 100 | 10 | 100 | 0 | 0 |
| | | 0±1 | 5±7 | 95±7 | 1±1 | 99±1 | 0±0 | 99±1 | 1±1 | 0±0 |
| JU2095 (1.79 × WT EGF, ectopic Notch) | | 0 | 6 | 94 | 1 | 99 | 0 | 100 | 0 | 0 |
| | | 1±1 | 8±10 | 91±10 | 2±2 | 98±2 | 0±0 | 99±1 | 1±1 | 0±0 |
| JU2092 (2.75 × WT EGF, ectopic Notch) | | 5 | 24 | 71 | 0 | 100 | 0 | 100 | 0 | 0 |
| | | 4±5 | 34±20 | 62±21 | 20±16 | 80±16 | 0±0 | 99±2 | 1±1 | 0±0 |
| **Reduced EGF** | | | | | | | | | | |
| CB1417 (EGF hypomorph) | | 0 | 0 | 100 | 1 | 99 | 0 | 54 | 0 | 46 |
| | | 0±0 | 1±1 | 99±1 | 2±2 | 44±20 | 54±21 | 55±20 | 10±7 | 35±22 |
| JU2095 (EGF hypomorph, ectopic Notch) | | 0 | 1 | 99 | 0 | 15 | 85 | 72 | 0 | 28 |
| | | 0±1 | 3±5 | 97±5 | 1±2 | 65±16 | 34±16 | 61±16 | 34±15 | 5±9 |

For each mutant, the experimental percentages (top) and simulated percentages (mean ± SD, bottom) for each VPC becoming each fate are shown. In the experimental data, fates for P4.p and P8.p (or P5.p and P7.p) have been averaged, since we assume that the pattern is symmetrical around the anchor cell.

change. Although our model is not capable of reproducing a small number of tertiary fated P5.p cells under the JU2113 perturbation, since the signaling profile moves them slightly outside the tristability region, it does reproduce the increase in primary fated P5.p, by approaching the cusp mid-line in the signal space while not leaving the bistability region inside the cusp (Table 2 and Fig 6A).

We also tested whether the model is able to reproduce mutants with ectopic Notch expression. Similar to the previous subset of data, these perturbations are silent unless crossed with EGF overexpression perturbations. As in [5], we modeled these perturbations as an additive constant for the Notch signaling parameter as, in this case, the perturbation is independent of the state of the cell and its neighbors (see S1 Appendix). We fitted this constant to the experimental outcome of the JU2092 mutant, where ectopic Notch is combined with a 1.25-fold EGF overexpression. This fitting resulted in a constant equal to 0.12. With this, we were able to fit the three crosses shown in Table 2. Interestingly, in the strongest EGF overexpression perturbation (mutant JU2092), P4.p was positioned close to the right of the intersection between the cusp mid-line and the fold line, giving a mix of tertiary and secondary fates, as observed in the data (Fig 6A).

Finally, we simulated EGF hypomorph mutants. Interestingly, these perturbations show that a strong decrease of EGF signals gives a mix of primary and tertiary fated P6.p cells while not affecting the WT pattern of the rest of VPCs. Moreover, a cross with a mutant with mild ectopic Notch activity, such as the one considered above, promotes primary fated P6.p,

showing again epistasis between the signals. To simulate these perturbations, we fitted a multiplicative downregulation of EGF signal to the EGF hypomorph data, CB1417. With a reduction of 0.36 × WT EGF, our model can reproduce the mix of tertiary and primary fated P6.p observed in the EGF hypomorph mutant (CB1417) (Fig 6C). This is achieved because a reduction in EGF positions the signaling profile of P6.p close to the fold line while still being to the left of the cusp-midline (Fig 6B). Simulating the cross of this perturbation with ectopic Notch (mutant JU2095) moves P6.p signaling profile further from the fold line, promoting vulval fates. However, it also positions it closer to the cusp mid-line and therefore, also promotes secondary fates in P6.p in our simulations (Fig 6B). The model is also not able to reproduce the high probability of P5.p staying in tertiary fate. We believe that the differences between simulations and experiments observed in P6.p could be fixed by adding the perturbations to the training data set. Reproducing the differences observed in P5.p would likely require a non-linear mapping of the fold into the signal space.

## Model predictions

Having reproduced a large set of data, we explored whether the model could predict interesting outcomes.

We tested whether crossing two silent perturbations in the training data could show a new epistatic event. Halving EGF ligand or Notch receptors does not affect the wild type phenotype (Table 1 (3–4)). In our model, this is because making either of these two perturbations leaves cells within the same stability regions of the fate map. However, our model suggests that, if both perturbations are crossed, we observe epistasis, where induction of P5.p to secondary fates is strongly reduced by almost half (Fig 7A). This is consistent with predictions shown in [3]. Biologically, this prediction suggests that EGF promotes secondary fates which, at first, could seem counterintuitive.

AC ablation experiments are very insightful, as they are the only experiments that provide information about the trajectory of the cells in the landscape. However, AC ablation data is only available under a WT signaling profile. As we mentioned earlier, a wild type pattern is achieved even under either half dose *lin-3* or *lin-12*. However, it is not known whether the pattern is dynamically formed in the same way as under WT signaling. We used our model to test this and observed an interesting effect.

Simulating AC ablation under a half dose of *lin-3*, the EGF ligand, showed that the process followed the same order as under WT signal, but proceeded more slowly (Fig 7B). Under this condition, our model predicts (1/3, 1/3, 1/3) probabilities for P6.p becoming one of the three fates if AC is ablated at a time between 3˚ divided stage and 2-cell stage. However, under a half dose of *lin-12*, a Notch receptor, our model predicted that there is very low probability of P6.p becoming a secondary fate under any ablation time, as observed in the WT and *lin-3* mutant (Fig 7C). Instead, there is a direct transition from tertiary to primary fate. This is also consistent with the effect observed in [3].

Finally, we explored whether there was a prediction that could highlight the differences between the model presented here and that proposed in [3, 5]. An important difference between the two models is the fact that the binary flip with cusp model is not symmetric, in that once cells have sufficiently left the basin of attraction of the tertiary fate they cannot go back for any value of the external signals. For example, if isolated cells (with no autocrine or paracrine Notch signaling) are exposed to a short EGF pulse of the same WT strength (from time $t_1 = 0$ to $t_2 = 0.4$), our model predicts that most of them would leave the basin of attraction of tertiary fate and commit to either the primary or secondary fate (S3(A) Fig). Moreover, due to the directions of the flow lines, there is no signaling profile that would be able to

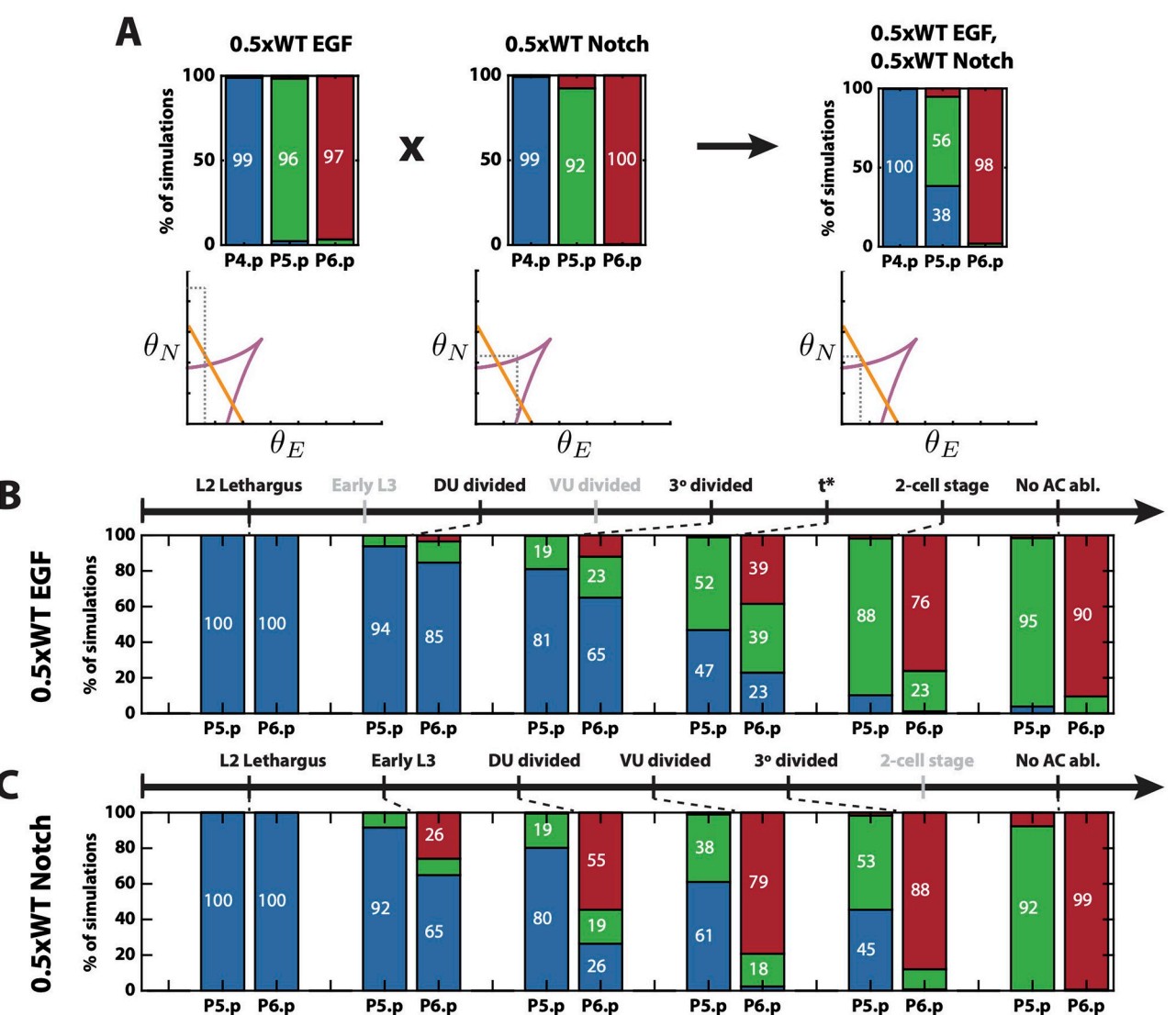

**Fig 7. Model's predictions.** (A) Predicted epistasis under the cross of reduced EGF ligand and Notch receptors perturbations. Top: Simulated proportions under the corresponding signaling profiles. Bottom: Schematics of the simulated positions of P5.p in the signal space under the different signaling profiles. (B) Predicted pattern proportions for different AC ablation times under a reduced EGF ligand perturbation. $t^*$ is a predicted time between 3° divided and 2-cell stages. The stages for which simulations are not shown are colored in gray. (C) Predicted pattern proportions for different AC ablation times under a reduced Notch receptors perturbation. The names of the stages for which simulations are not shown are colored in gray.

return cells back to tertiary fate. For example, if this short EGF pulse is followed by a very short Notch signaling pulse ($t_2 = 0.4$ to $t_3 = 0.62$), cells remain outside the basin of attraction of the tertiary fate and would differentiate into the secondary fate (S3(A) Fig). However, this is not the case in the model proposed by [3, 5]. As in our model, isolated cells exposed to the same short EGF pulse leave the tertiary fate and commit to primary fate, resulting in 90% of primary-fated cells (S3(B) Fig). Strikingly, if this is followed by the short Notch signaling pulse, the cells return to the basin of attraction corresponding to the tertiary fate, resulting in 71% now differentiating to the tertiary fate and only 19% differentiating to the primary fate (S3(B) Fig). Performing this or a similar experiment would be able to differentiate between the two models.

## Discussion

Here we have presented a method to build simple landscapes starting from qualitative observations, which can then be fitted to a large amount of quantitative data. With this method we have shown that a very simple three-way landscape is able to reproduce the complex vulval patterning process in *C. elegans*. Given the number of stable cell fates observed in the data, CT is a powerful theory to classify and build landscapes with the desired characteristics. By coupling it with an efficient parameter fitting method, such as the one presented based on ABC SMC, one can check which proposed models are consistent with the data and choose the simplest one.

An important difference between our model and the one developed in [3, 5] is that while in their model all states are equivalent, giving a three-fold symmetric potential, in our model, the cells go through two cell state transitions: a first decision between vulval and non-vulval fates, and then, for cells that adopt vulval fates, a decision between primary and secondary. This may be a general landscape which can be used whenever a cell first decides whether to leave a precursor state or not, and then, once leaving that state, decides between a pair of options.

Here we have taken advantage of vulval development in *C. elegans* to illustrate this new mathematical framework, however, this is a very complex problem involving patterning of several cells that interact through both external and paracrine signals, and therefore, the state of each cell depends upon that of its neighbors. Moreover, we lacked detailed information about cell fate and signal dynamics in time, as data is only available on the final phenotypes. The only data available on dynamic perturbations were those in which the AC was ablated and even in this case, again the only available phenotype was the final one and the specific ablation times were not available. This resulted in relatively large number of unknowns and corresponding simplifying assumptions, such as the model for EGF increase in time, which complicated the model and the analysis. This approach, however, is ideal for modeling single cell data for which one is able to observe cell transitions along the protocol time as well as controlling signaling in time, as we show in [36].

An advantage of using this approach, where the model is built from basic transitions, is that the system can be easily evolved. Here we have postulated that the mapping between signals and control parameters was linear, and we were able to reproduce a great amount of data. However, we also showed that the model struggled to reproduce some details observed in the EGF hypomorph perturbation. This could suggest that a more sophisticated mapping would be needed in this case, and with our approach, it would be possible to locally modify the mapping in the low EGF region without changing the remainder of the model.

With the fast development of experimental techniques to observe single cells, such as single cell RNA-seq together with live signaling and cell fate reporters, vast amounts of data are becoming available. Building gene regulatory networks that account for all the subtleties observed in the data proves to be challenging. In fact, there is work trying to connect GRNs and evolution to quasi-potential landscape models [37–43], and [44] shows that looking at the GRN through the lens of landscape models can be a useful way to classify standard behaviors. Moreover, we can expect that the dynamics of any GRN will be largely described by the normal forms that Catastrophe Theory and Dynamical Systems Theory provide, as they develop by successive transitions among a small panel of fates [25, 45–47]. Therefore, the framework proposed here, that centers on the essence of the process without the complicating molecular details, can alleviate these challenges and, in fact, could benefit from this new information. This approach to building landscape models has several advantages in comparison to GRN models. First, as mentioned before, since it focuses on the essence of the process rather than on the mechanistic details, one can build a landscape model from simple qualitative data about the state transitions. Secondly, while there are many network structures that can reproduce

similar observations, there are only a few distinct landscape topologies with a given number of attractors. We believe there are many exciting applications of this framework to more cell differentiation processes.

## Supporting information

**S1 Appendix. Supporting information.** In this note we introduce concepts from Dynamical Systems and Catastrophe Theory, give further details about the mathematical underpinnings of the methodology to create the landscape model as well as about the implementation of the ABC SMC algorithm and fitting results discussed in the main paper.
(PDF)

**S1 Fig. Evolution of the distributions of the parameters in the fitting of the training data.** Histograms and two dimensional scatter plots of the $N = 2 \times 10^4$ particles sampled from the approximated posterior distribution given the training data in Table 1 at the first (blue) and last (red) steps of the algorithm.
(TIF)

**S2 Fig. Further analysis of the posterior distributions of the parameters.** (A) Evolution of the variance of the parameters at each step of the ABC algorithm, normalised by the variance of the prior. (B) Correlation matrix of the parameters at the last step of the algorithm. (C) Approximated posterior distributions of the parameters defining the linear transformation of the signal space into the control space.
(TIF)

**S3 Fig. The trajectories of isolated cells show differences between the binary flip with cusp landscape model and the model proposed in [3, 5].** (A) Simulations of the differentiation of isolated VPCs exposed to EGF from time $t_1 = 0$ to $t_2 = 0.4$ followed by Notch from $t_2 = 0.4$ to $t_3 = 1$ following the binary flip with cusp model, plotted on the landscape corresponding to the absence of signaling. The mean trajectory of VPCs is represented as a white line. The distribution of cell states at time $t = 0.4$ is shown as gray dots. The distribution of cell states at time $t = 0.62$ is shown as white dots. (B) Simulations of the differentiation of isolated VPCs exposed to EGF from time $t_1 = 0$ to $t_2 = 0.4$ followed by Notch from $t_2 = 0.4$ to $t_3 = 1$ following the model proposed in [3, 5], plotted on the landscape corresponding to the absence of signaling. The mean trajectory of VPCs is represented as a white line. The distribution of cell states at time $t = 0.4$ is shown as gray dots. The distribution of cell states at time $t = 0.62$ is shown as white dots. Attractors and saddles are represented by black dots and triangles, respectively. Basins of attraction are colored blue, green or red if they represent tertiary, secondary or primary fates, respectively.
(TIF)

**S1 Video. Model dynamics of a simulated WT pattern.** Trajectories of 150 simulated P4–6.p in the phase space (top) and their signaling profiles in the signal space (bottom). The trajectories of each of the 150 simulations are shown as faded gray dots, and we choose one of them (in darker gray and white boundary line) to show its corresponding landscape changes according to its signaling profiles. Basins of attraction are colored blue, green or red if they represent tertiary, secondary or primary fates, respectively.
(MP4)

**S2 Video. Model dynamics of a simulated *let −23* mosaic mutant.** Trajectories of 150 simulated P4–6.p in the phase space (top) and their signaling profiles in the signal space (bottom). The trajectories of each of the 150 simulations are shown as faded gray dots, and we choose

one of them (in darker gray and white boundary line) to show its corresponding landscape changes according to its signaling profiles. Basins of attraction are colored blue, green or red if they represent tertiary, secondary or primary fates, respectively.
(MP4)

**S3 Video. Model dynamics of a simulated half dose *lin −3* mutant.** Trajectories of 150 simulated P4–6.p in the phase space (top) and their signaling profiles in the signal space (bottom). The trajectories of each of the 150 simulations are shown as faded gray dots, and we choose one of them (in darker gray and white boundary line) to show its corresponding landscape changes according to its signaling profiles. Basins of attraction are colored blue, green or red if they represent tertiary, secondary or primary fates, respectively.
(MP4)

**S4 Video. Model dynamics of a simulated half dose *lin −12* mutant.** Trajectories of 150 simulated P4–6.p in the phase space (top) and their signaling profiles in the signal space (bottom). The trajectories of each of the 150 simulations are shown as faded gray dots, and we choose one of them (in darker gray and white boundary line) to show its corresponding landscape changes according to its signaling profiles. Basins of attraction are colored blue, green or red if they represent tertiary, secondary or primary fates, respectively.
(MP4)

**S5 Video. Model dynamics of a simulated Notch null, 2ACs mutant.** Trajectories of 150 simulated P4–6.p in the phase space (top) and their signaling profiles in the signal space (bottom). The trajectories of each of the 150 simulations are shown as faded gray dots, and we choose one of them (in darker gray and white boundary line) to show its corresponding landscape changes according to its signaling profiles. Basins of attraction are colored blue, green or red if they represent tertiary, secondary or primary fates, respectively.
(MP4)

**S6 Video. Model dynamics of a simulated mutant with no Notch signaling and WT EGF.** Trajectories of 150 simulated P4–6.p in the phase space (top) and their signaling profiles in the signal space (bottom). The trajectories of each of the 150 simulations are shown as faded gray dots, and we choose one of them (in darker gray and white boundary line) to show its corresponding landscape changes according to its signaling profiles. Basins of attraction are colored blue, green or red if they represent tertiary, secondary or primary fates, respectively.
(MP4)

**S7 Video. Model dynamics of a simulated EGF overexpression mutant.** Trajectories of 150 simulated P4–6.p in the phase space (top) and their signaling profiles in the signal space (bottom). The trajectories of each of the 150 simulations are shown as faded gray dots, and we choose one of them (in darker gray and white boundary line) to show its corresponding landscape changes according to its signaling profiles. Basins of attraction are colored blue, green or red if they represent tertiary, secondary or primary fates, respectively.
(MP4)

**S8 Video. Model dynamics of a simulated mutant where the AC was ablated at the L2 lethargus stage.** Trajectories of 150 simulated P4–6.p in the phase space (top) and their signaling profiles in the signal space (bottom). The trajectories of each of the 150 simulations are shown as faded gray dots, and we choose one of them (in darker gray and white boundary

line) to show its corresponding landscape changes according to its signaling profiles. Basins of attraction are colored blue, green or red if they represent tertiary, secondary or primary fates, respectively.
(MP4)

**S9 Video. Model dynamics of a simulated mutant where the AC was ablated at the early L3 stage.** Trajectories of 150 simulated P4–6.p in the phase space (top) and their signaling profiles in the signal space (bottom). The trajectories of each of the 150 simulations are shown as faded gray dots, and we choose one of them (in darker gray and white boundary line) to show its corresponding landscape changes according to its signaling profiles. Basins of attraction are colored blue, green or red if they represent tertiary, secondary or primary fates, respectively.
(MP4)

**S10 Video. Model dynamics of a simulated mutant where the AC was ablated at the DU divided stage.** Trajectories of 150 simulated P4–6.p in the phase space (top) and their signaling profiles in the signal space (bottom). The trajectories of each of the 150 simulations are shown as faded gray dots, and we choose one of them (in darker gray and white boundary line) to show its corresponding landscape changes according to its signaling profiles. Basins of attraction are colored blue, green or red if they represent tertiary, secondary or primary fates, respectively.
(MP4)

**S11 Video. Model dynamics of a simulated mutant where the AC was ablated at the VU divided stage.** Trajectories of 150 simulated P4–6.p in the phase space (top) and their signaling profiles in the signal space (bottom). The trajectories of each of the 150 simulations are shown as faded gray dots, and we choose one of them (in darker gray and white boundary line) to show its corresponding landscape changes according to its signaling profiles. Basins of attraction are colored blue, green or red if they represent tertiary, secondary or primary fates, respectively.
(MP4)

**S12 Video. Model dynamics of a simulated mutant where the AC was ablated at the 3˚ divided stage.** Trajectories of 150 simulated P4–6.p in the phase space (top) and their signalling profiles in the signal space (bottom). The trajectories of each of the 150 simulations are shown as faded gray dots, and we choose one of them (in darker gray and white boundary line) to show its corresponding landscape changes according to its signaling profiles. Basins of attraction are colored blue, green or red if they represent tertiary, secondary or primary fates, respectively.
(MP4)

**S13 Video. Model dynamics of a simulated mutant where the AC was ablated at the 2-cell stage.** Trajectories of 150 simulated P4–6.p in the phase space (top) and their signalling profiles in the signal space (bottom). The trajectories of each of the 150 simulations are shown as faded gray dots, and we choose one of them (in darker gray and white boundary line) to show its corresponding landscape changes according to its signalling profiles. Basins of attraction are colored blue, green or red if they represent tertiary, secondary or primary fates, respectively.
(MP4)

## Acknowledgments

We thank Eric D. Siggia and Francis Corson for extensive conversations and advice; Paul Brown at the University of Warwick and the Center for Research Computing at Rice

University for their technical assistance in running parallel computations on the clusters; and Miguel Ángel Ortiz Salazar for his feedback on the manuscript.

## Author Contributions

**Conceptualization:** Elena Camacho-Aguilar, David A. Rand.

**Data curation:** Elena Camacho-Aguilar.

**Formal analysis:** Elena Camacho-Aguilar, David A. Rand.

**Funding acquisition:** Aryeh Warmflash, David A. Rand.

**Investigation:** Elena Camacho-Aguilar, David A. Rand.

**Methodology:** Elena Camacho-Aguilar, Aryeh Warmflash, David A. Rand.

**Project administration:** David A. Rand.

**Resources:** Aryeh Warmflash, David A. Rand.

**Software:** Elena Camacho-Aguilar.

**Supervision:** Aryeh Warmflash, David A. Rand.

**Validation:** Elena Camacho-Aguilar, David A. Rand.

**Visualization:** Elena Camacho-Aguilar, Aryeh Warmflash, David A. Rand.

**Writing – original draft:** Elena Camacho-Aguilar, David A. Rand.

**Writing – review & editing:** Elena Camacho-Aguilar, Aryeh Warmflash, David A. Rand.

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
