## [Decision Letter · Decision Letter 0]

24 Mar 2021

Dear Dr. Camacho Aguilar,

Thank you very much for submitting your manuscript "Quantifying cell transitions in C. elegans with data-fitted landscape models" for consideration at PLOS Computational Biology.

As with all papers reviewed by the journal, your manuscript was reviewed by members of the editorial board and by several independent reviewers. In light of the positive reviews (below this email), we would like to invite the resubmission of a revised version that takes into account the reviewers' comments.

As you can see, all reviews are very positive about the work. They ask for some clarifications and have a few remaining questions. Please note that review #2 is in an attached Word file.

We cannot make a final decision about publication until we have seen the revised manuscript and your response to the reviewers' comments. Your revised manuscript may be sent to reviewers for further evaluation.

Sincerely,

Roeland M.H. Merks, Ph.D

Associate Editor

PLOS Computational Biology

Mark Alber

Deputy Editor

PLOS Computational Biology

Reviewer's Responses to Questions

**Comments to the Authors:**

Reviewer #1: In short, I consider this to be an outstanding paper. The authors combine ideas from catastrophe theory — an area that has been around a fair while, but which has rarely been employed to understand real systems beyond simply pointing out similarities in a rather hand-waving fashion — with (approximate) Bayesian inference employing real data. They employ this to a classical developmental model system, the vulva development in C elegans.

It builds on and significantly improves upon recent work which established more formal analysis of Waddington’s epigenetic landscape and developmental systems.

I did like the systematic development of a model that encapsulates the desired cell fate transitions/bifurcations: it was remarkably clear to follow and each step is explained with sufficient detail to be reproducible.

Model construction is well described and the authors then attempt to calibrate the model against data using ABC. The approach taken by the authors is again clearly motivated and explained. One question that I would like to be addressed more clearly is the choice of distance function, Eqn. (10). The second term is well explained, but in comparing the probabilities, other alternative distances, in particular information theoretical distances could have been considered? Such a discussion could go into Appendix 2.

The way in which the model is calibrated against the data is well explained in Figure 4. But I was intrigued by the results of 4B and would like to learn more. The videos accompanying

Figure 4 are superb by the way.

From S 4.1 I was also wondering if the authors could provide some indication as to which parameters can be inferred (or are constrained by the data — I also fear that the text is a bit too small in the current resolution).

In the validation section the authors show that their model can capture epistatic effects. This is an additional appeal of the approach developed by the authors. Figure 6A shows this, but I found the discussion somewhat terse.

A central statement in the discussion is “Moreover, we can expect that the dynamics of any GRN will be largely described by the normal forms that Catastrophe Theory and Dynamical Systems Theory provide.” I think I would on balance agree with this fundamentally; but I wish the authors would try to make this point in more detail.

I would also suggest that the authors make a more obvious statement that the methodology is available at a dedicated github site (it is only mentioned in the appendices and in the pdf not close to any mention of “software” or “methodology”. Finally, some indication as to the run-time would be helpful.

##Minor Comments:

- Line 214 & 223 and throughout: be clearer with your notation x, y, y_1, y_i. What is the range of index i?

- Line 593: “article to be submitted” is not a suitable reference.

I waive my right to anonymity,

Michael Stumpf

Reviewer #2: The review has been uploaded as an attachment

Reviewer #3: This is a brilliant paper. It should be published without further changes.

The study of cell fate and differentiation is an ongoing challenge in cell biology. For mathematical modellers it is natural to think of cell states as attractors of a dynamical system, and hence dynamical systems models have been prevalent. In systems biology, models of gene regulatory network that participate in cell fate decisions are widely used. However, systems biology models tend to suffer from overwhelmingly high dimensionality. This high dimensionality seems inevitable, since genomes contain thousands of genes and transcription factors. In this groundbreaking paper, Camacho-Aguilar et al. propose a new method to construct comparatively low-dimensional dynamical systems that can be quantitatively fitted to experimental data. They very convincingly illustrate the utility of the method by applying it to cell state transitions during C. elegans vulval development. Their model can fit a range of results on the rates of cell state transitions under varying experimental conditions, and correctly predicts further results without fitting. The authors further illustrate the utility of their method by predicting outcomes of a few potential future experiments.

I believe the paper makes a big conceptual advance that will be very important to the quantitative study of cell state transitions in many different systems. It really moves the field forward. In addition, it provides new insights C. elegans development. Hence, I strongly recommend the paper for publication.

The paper is very thorough and well written. I particularly like that it introduces relevant concepts on catastrophe theory from scratch, and that simulation codes are available for download. The only comment that I have relates to previous attempts at constructing Waddington-like landscapes. The authors cite some of these. However, it seems they have missed some of the related work by Yogi Jaeger and colleagues, such as (I am not one of the authors): https://bmcsystbiol.biomedcentral.com/articles/10.1186/1752-0509-8-43

I would like to apologise to the authors for the delay in submitting this review.

**Have all data underlying the figures and results presented in the manuscript been provided?**

Reviewer #1: Yes

Reviewer #2: Yes

Reviewer #3: Yes

PLOS authors have the option to publish the peer review history of their article (what does this mean?). If published, this will include your full peer review and any attached files.

Reviewer #1: **Yes: **Michael P.H. Stumpf

Reviewer #2: **Yes: **Anna Alemany

Reviewer #3: No
---

## [Decision Letter · Decision Letter 1]

3 May 2021

Dear Dr. Camacho Aguilar,

We are pleased to inform you that your manuscript 'Quantifying cell transitions in C. elegans with data-fitted landscape models' has been provisionally accepted for publication in PLOS Computational Biology.

Best regards,

Roeland M.H. Merks, Ph.D

Associate Editor

PLOS Computational Biology

Mark Alber

Deputy Editor

PLOS Computational Biology

Reviewer's Responses to Questions

**Comments to the Authors:**

Reviewer #1: The authors have addressed all my points to my complete satisfaction.

This is a very good piece of work, and superbly presented.

Michael Stumpf

Reviewer #2: The authors addressed all my comments constructively.

I would like to take this opportunity to congratulate the authors for this work, which will surely move forward the field of computationa biology.

Reviewer #3: The authors have addressed my comment and I recommend this paper for publication in Plos Computational Biology.

**Have the authors made all data and (if applicable) computational code underlying the findings in their manuscript fully available?**

Reviewer #1: Yes

Reviewer #2: Yes

Reviewer #3: Yes

PLOS authors have the option to publish the peer review history of their article (what does this mean?). If published, this will include your full peer review and any attached files.

Reviewer #1: **Yes: **Michael P.H. Stumpf

Reviewer #2: **Yes: **Anna Alemany

Reviewer #3: No

---

## [Editor Report · Acceptance letter]

25 May 2021

PCOMPBIOL-D-21-00288R1 

Quantifying cell transitions in *C. elegans* with data-fitted landscape models

Dear Dr Camacho Aguilar,

I am pleased to inform you that your manuscript has been formally accepted for publication in PLOS Computational Biology. Your manuscript is now with our production department and you will be notified of the publication date in due course.

With kind regards,

Olena Szabo
